# Accounting for albedo in carbon market protocols

**Lynn M. Riley** [1,2] ✉, **Susan C. Cook-Patton** [3,4], **Loren P. Albert** [2], **Christopher J. Still** [2], **Christopher A. Williams** [5] & **Jacob J. Bukoski** [2]

The climate benefits of some Voluntary Carbon Market projects may be overestimated due to a lack of accounting for albedo change. Here we analyze 172 Afforestation, Reforestation, and Revegetation projects within the market and find more than 10% occur in places where albedo may entirely negate the climate mitigation benefit, and a quarter occur in places where albedo may halve the mitigation benefit. Yet, the majority are concentrated where albedo changes are expected to be minimal, and 9% of projects occur where albedo would augment the mitigation benefit. Recent data are making albedo accounting possible, and we outline an iterative approach for incorporating albedo considerations into carbon crediting protocols to prioritize projects with greater climate benefit and more accurately quantify credits that may be used to address unabated emissions. We also call on the scientific community to create tools to enable accounting for other important biophysical changes, such as evapotranspiration, which is not yet quantifiable within the Voluntary Carbon Market.

Natural climate solutions (NCS)—conservation, improved management, and restoration actions that result in climate benefits—can provide cost-effective climate mitigation[1,2] and are essential to achieving global climate goals[3,4]. Carbon markets are a prominent mechanism for funding NCS, providing marketplaces for trading verified metric tons of carbon dioxide equivalent ($CO_2e$) emissions that have been avoided or removed (i.e., carbon credits) from the atmosphere. Carbon credits are produced through established protocols that aim to ensure each credit corresponds to an actual climate benefit. Although NCS can be employed for a variety of goals, use of NCS to abate emissions requires very high standards to ensure reliability and durability[5,6]. To date, carbon crediting protocols account for the biochemical (i.e., greenhouse gas) benefits of projects, but largely omit the biophysical (non-greenhouse gas) factors that can also substantially influence the global climate[6–9].

One biophysical factor that has received renewed attention is surface albedo, or the amount of shortwave radiation reflected to space relative to what is absorbed by the surface and converted to heat

in the Earth system (Fig. 1). Changes in the albedo of Earth's surface can substantially contribute to the net climate impact of forestation projects, and are a well-known phenomenon that predates recent interests in scaling NCS[10–12]. Specifically, while reforestation sequesters atmospheric $CO_2$ through forest growth, it also tends to decrease surface albedo, which can partially (or fully) cancel out the climate benefit of $CO_2$ removal[9,12] (Fig. 1). This offsetting behavior has most commonly been described for boreal regions of the globe[10], which has led to general recommendations to prioritize reforestation of equatorial regions over northern latitudes[13]. However, recent advances in remote sensing and geostatistical modeling are now providing improved and spatially-resolved assessments of albedo shifts from forestation, prompting additional research and innovation in the space[14,15].

Ignoring albedo in NCS projects can lead to inflated estimates of climate mitigation or even projects that exacerbate rather than mitigate climate change[14,16]. Although albedo has been difficult to account for, recent, independently-derived and open-source maps provide spatially-explicit albedo changes expected from reforestation[14]. The

¹American Forest Foundation, Washington, DC, USA. ²Department of Forest Ecosystems & Society, Oregon State University, Corvallis, OR, USA. ³Tackle Climate Change Program, The Nature Conservancy, Arlington, VA, USA. ⁴Smithsonian Environmental Research Center, Edgewater, MD, USA. ⁵Graduate School of Geography, Clark University, Worcester, MA, USA. ✉e-mail: lriley@forestfoundation.org

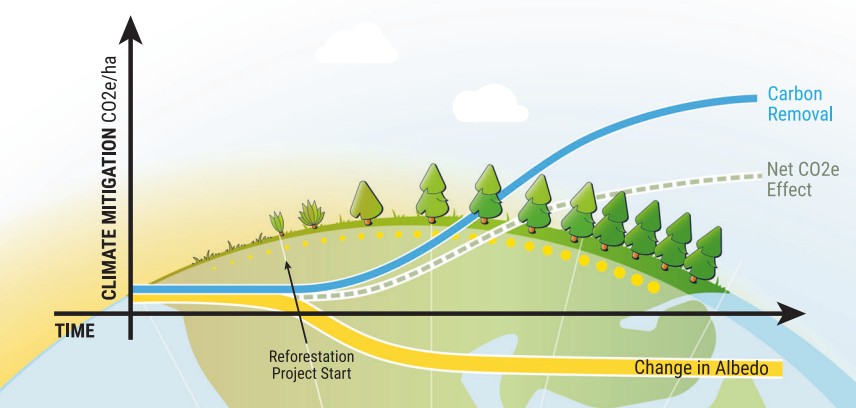

**Fig. 1 | Conceptual illustration of reforestation with a partial albedo deduction.** (partial negation of $CO_2$ removal climate benefit). Surface albedo declines after the project starts as the land surface becomes less reflective. As the forest grows, $CO_2$ is removed from the atmosphere. The net carbon dioxide equivalent ($CO_2$e) effect is the net of $CO_2$ removal and albedo change. As shown, the albedo deduction is expected to reach its maximum effect before $CO_2$ removal reaches its maximum effect. Conceptual illustration of reforestation with a partial albedo deduction © 2024 by Vin Reed is licensed under CC BY 4.0. To view a copy of this license, visit https://creativecommons.org/licenses/by/4.0/.

data estimate the expected albedo change in $CO_2$e for a suite of non-forest to forest land cover transitions ("single transition maps"), as well as the percentage of maximum $CO_2$ removal[17] that would be negated (or increased) by changes in albedo for the "most likely" reforestation transition (hereafter, the "Hasler data"). We refer to the latter as the "albedo deduction" (or "albedo benefit"). For example, a 50% albedo deduction in the Hasler data indicates that 50% of the maximum $CO_2$ removal from reforestation[17] is negated by albedo decreases expected from the most likely reforestation transition. Conversely, reforestation can create an albedo benefit by brightening the surface in some situations, such as when darker shrubs are reforested to woody savannas. The resulting net cooling is then greater than what would occur based on $CO_2$ removal alone.

Despite the potential for albedo benefits, albedo deductions are much more common due to the generally low albedo of forests relative to most other land cover types. Albedo deductions may even occur in tropical or temperate biomes, where forestation projects have generally been encouraged[13]. While the potential for albedo to alter the climate benefits of forest cover expansion is well known, we lack understanding of the impacts on existing NCS projects. Additionally, proposals for how albedo changes could be integrated into protocols for carbon market accounting are being actively discussed within the literature[8,18]. While recent work has quantified shifts in albedo from forest expansion at national scales[15], no such analysis has been performed for projects on the Voluntary Carbon Market (VCM), which seek to directly finance their activities by quantifying their climate impacts.

Here, we quantify the potential impact of albedo on 172 existing Afforestation, Reforestation, and Revegetation (ARR) projects and present an iterative approach for incorporating albedo into project prioritization, design, and accounting. To do so, we coupled 500-m resolution Hasler data with publicly available data on the location and expected carbon credit production of ARR projects in the VCM. We focused on ARR over other types of NCS for four reasons. First, we see increasing interest in $CO_2$ removal credits (which ARR projects produce)[19,20], including from net zero standards[21] and carbon credit buyer coalitions[22], as an essential negative emission technology as overshoot becomes more likely[8,23]. Second, datasets that quantify albedo effects from ARR are now available (e.g., Hasler data). Third, albedo change from ARR is expected to be consequential[24]. Fourth, lack of ARR albedo accounting has been noted throughout the scientific literature and has yet to be addressed[8,9,11,19,25], despite being both scientifically justified and feasible to measure[14,26].

## Results

We identified accessible geospatial and project documentation for 172 ARR projects across the VCM, which were distributed across five continents (Fig. 2). These projects are projected to provide nearly 800 million metric tons $CO_2$e of climate mitigation over the next century. None of the five registry standards and accompanying protocols used by these projects (Supplementary Table 2) account for changes in albedo, though one standard, the Verified Carbon Standard, signaled interest in biophysical accounting in the future[27].

### Albedo deductions for existing ARR projects
We find that the median albedo deduction across all projects is 18%. A quarter of projects are expected to face albedo deductions of 50% or more, including 12% where the entire $CO_2$ benefit is negated by changes in albedo. In contrast, 9% of projects are estimated to have an albedo benefit (Fig. 2). We generally found albedo deductions to be greater in biomes known to have greater risks of albedo impacts, such as deserts and arid ecosystems, woodlands, and temperate forests (Fig. 3). However, projects with the largest albedo deductions also occurred in a variety of contexts: on four continents (Africa, Asia, Australia, and South America), within 12 biomes, across project start dates (from decades-old projects to projects that began development in the past two years), across seven protocols, and across a range of geographic areas (from hundreds to hundreds of thousands of hectares and with varying distance from the equator) (Fig. 3; Table 1, Supplementary Tables 1 and 2). Most ARR projects occur in tropical and subtropical moist broadleaf forests where the albedo deduction is moderate (12% median). However, other biomes like montane grasslands also have many projects and much higher albedo deductions (39% median) (Supplementary Table 1).

### Albedo benefits
While projects with albedo deductions occurred across diverse contexts, we find that the 16 projects (9% of those assessed) with albedo benefits were more concentrated. These projects were all within Central America, South America, and Africa, and nearly all were within three tropical and subtropical biomes: Tropical and Subtropical Dry Broadleaf Forests; Tropical and Subtropical Grasslands, Savannas and Shrublands; and Tropical and Subtropical Moist Broadleaf Forests. While actual albedo benefits will depend on project specifics (such as the specific pre- and post-land cover achieved), these patterns suggest an opportunity to prioritize ARR projects within these biomes to enhance climate mitigation beyond carbon alone.

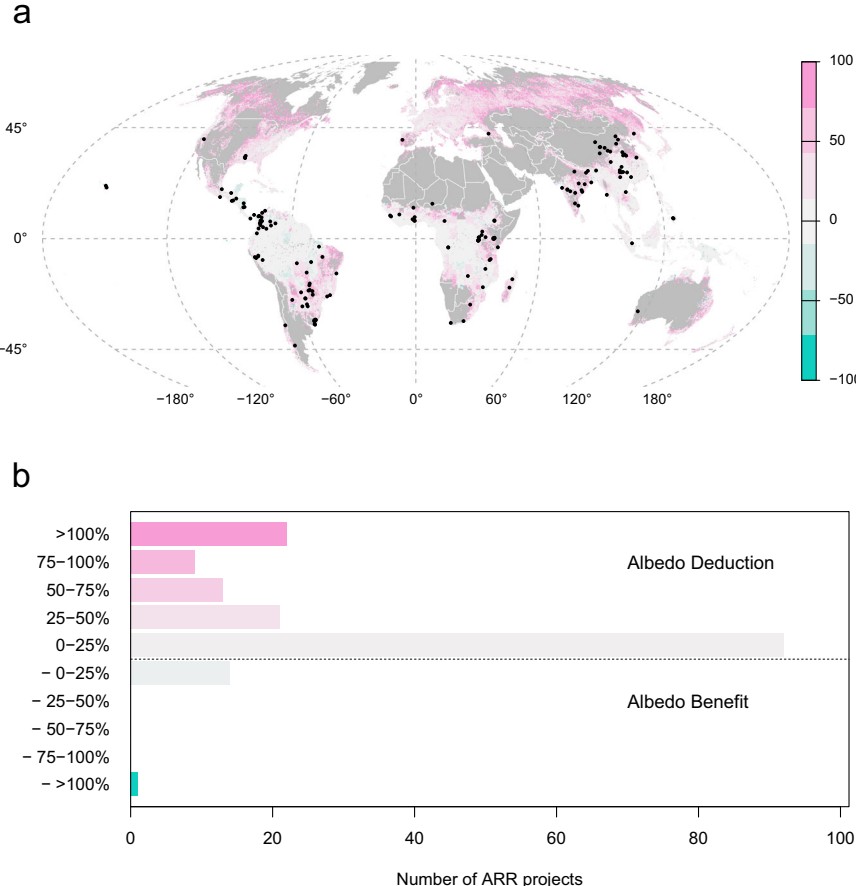

**Fig. 2 | Albedo deductions and benefits by location and project volume.**
**a** Locations of 172 projects assessed (black points) overlaid on the albedo deduction/benefit layer from the Hasler data. Pink indicates the level of albedo deduction (%), and blue indicates areas of albedo benefits (%). Gray indicates areas without data, typically places where forests are not suitable (e.g., deserts and grasslands). **b** The 172 projects are grouped by their median albedo change, where an albedo benefit indicates a greater net climate benefit from the project than calculated from $CO_2$ accounting alone, and an albedo deduction indicates a lower or negated climate benefit than calculated from $CO_2$ accounting alone.

## Effect of albedo deductions on carbon credits

Our assessment suggests that albedo changes are relevant for many VCM projects and that a quarter of projects produce half (or less) of the climate mitigation expected. Twenty-one projects, representing 30% of projected credits, may even exacerbate climate change. Based on this, there is a strong need to account for albedo in VCM protocols, especially when credits are sold to address emissions elsewhere. To support continuous and iterative improvement within the VCM[5], we developed a tiered approach that progressively integrates albedo considerations into VCM accounting. Additionally, we depict the impact of each tier on the 172 projects we assessed (Table 1 and Fig. 4), asking, what would the albedo-informed climate mitigation benefit be relative to today's VCM accounting? This tiered framework is designed to facilitate a phased transition from an approach that addresses the albedo accounting gap today (Tier 1); to one that offers near-term, transparent, and reproducible carbon accounting protocol improvements (Tier 2); and ultimately to one that identifies opportunities for more intensive accounting of albedo changes, but that may require additional research, communication, data, and/or tools (Tiers 3 and 4). In this way, incremental improvements aligned with growing awareness, trust, and scientific innovation can address the market need.

For Tier 1, we propose that new projects incorporate albedo by prioritizing project sites or forest types where albedo is anticipated to have a lower deduction or a benefit. Project funders could ask about this in due diligence processes. Project ratings agencies could add it to their assessment methodologies. The Hasler data allows projects to

demonstrate that they are in areas with lower potential albedo deductions. As this would not impact existing projects (only future projects), the projected credits for this tier match what is reported by projects today.

For Tier 2, albedo could be incorporated into protocols' "no net harm" guidelines for environmental impacts using the Hasler maps and data. Where an albedo deduction is large, additional follow-up could ensure that, when assessed at higher resolution, the effect was negligible. Where this cannot be demonstrated, the project could be deemed ineligible if the albedo deduction exceeds a specified threshold. We present 50% and 100% thresholds (i.e., negating half or all of the climate benefits from carbon sequestration) as examples, but others are possible. For the 172 projects we assessed, a 100% albedo deduction would render 21 (12%) projects ineligible, lowering the total projected credit production by 30.0% (Table 1 and Fig. 4). A 50% albedo deduction threshold would make 43 projects (25%) ineligible, lowering the total projected credit production by 39.1% (Table 1 and Fig. 4). We note that a relatively small reduction in eligible projects would lead to a relatively larger reduction in projected credits (e.g., 25% of projects ineligible versus 39.1% of credits ineligible in Tier 2 with a 50% threshold). This suggests that the projects with the largest albedo deductions appear to also be producing the most credits, perhaps due to being of larger geographic size.

For Tier 3, eligible projects could be required to discount total credit production by anticipated albedo change. The Hasler data provides multiple maps that could facilitate this. For example, projects

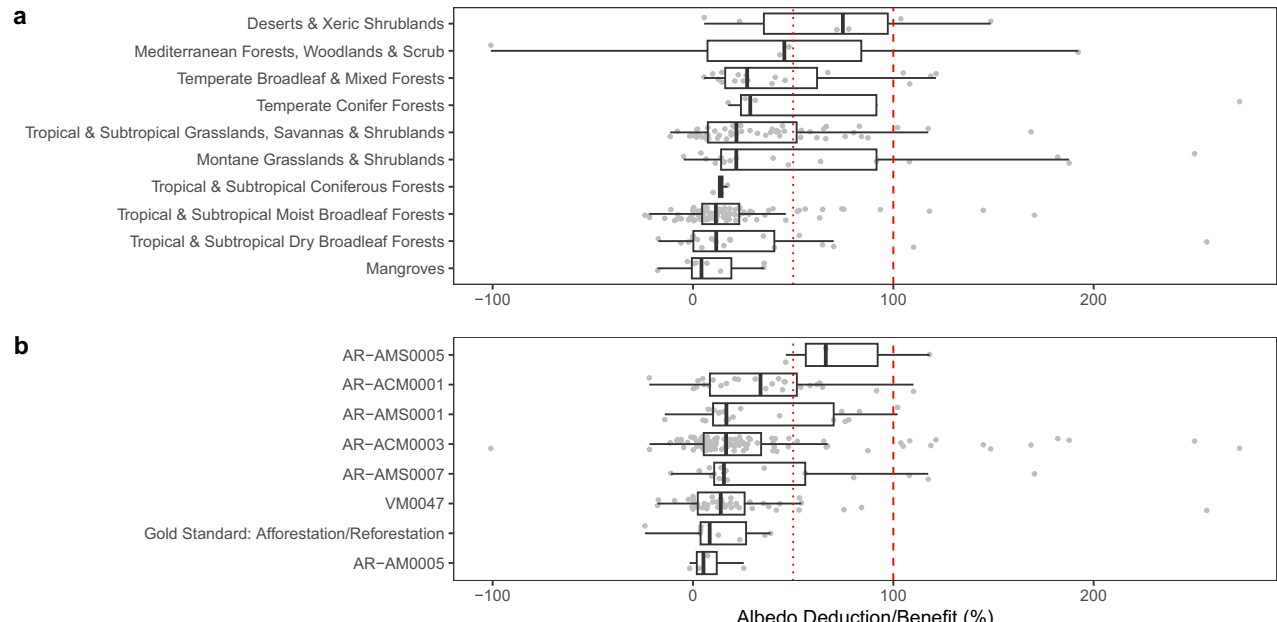

**Fig. 3 | Albedo deductions/benefits across the ARR projects assessed.** Albedo deductions and benefits of the projects assessed are summarized by (**a**). biomes and (**b**). protocols. The protocols listed (e.g., AR-AMS0005, etc.) in **b** represent the names of the carbon accounting methodologies used by the projects; links to these protocols are available in Supplementary Table 2. Albedo deductions beyond 300%, which accounted for 6% of projects, are excluded from this figure to improve readability. Biomes and protocols with at least three projects are included. Positive values indicate places where there is an albedo deduction to the climate benefit; negative values indicate places where there is no albedo deduction but rather an increase in climate mitigation due to changes in albedo (an albedo benefit). Dotted red line indicates 50% albedo deduction, and dashed red line indicates 100% albedo deduction (i.e., full negation of climate benefits from estimated ARR carbon sequestration).

**Table 1 | Impacts of iterative tiers to account for albedo deductions and benefits**

| Tier | Ineligibility threshold | Discounting | Benefits | Projected credits with albedo accounting (million metric tons $CO_2$) | Difference from current projected credits |
|---|---|---|---|---|---|
| 1 | -- | -- | -- | 795 | N/A |
| 2 | 100% Median Albedo | -- | -- | 556 | 30.0% fewer credits |
|   | 50% Median Albedo | -- | -- | 484 | 39.1% fewer credits |
| 3 | 100% Median Albedo | All other albedo deductions | -- | 425 | 46.5% fewer credits |
|   | 50% Median Albedo | All other albedo deductions | -- | 405 | 49.0% fewer credits |
| 4 | 100% Median Albedo | All other albedo deductions | Albedo benefits accounted | 429 | 46.0% fewer credits |
|   | 50% Median Albedo | All other albedo deductions | Albedo benefits accounted | 408 | 48.6% fewer credits |

Tier 1 represents the current projected credits, in million metric tons $CO_2$, expected across all 172 projects assessed from self-reported projections found within each project's project description. Tier 2 introduces a threshold of albedo deduction beyond which projects would be ineligible (an "ineligibility threshold"), and shows two potential scenarios of ineligibility thresholds: 50% and 100%. These thresholds refer to places where there is a 50% or 100% albedo deduction to the climate benefit. Tier 3 builds on Tier 2. After removing ineligible projects based on the ineligibility threshold, Tier 3 also discounts credits in the remaining eligible projects by their albedo deduction. Tier 4 builds on Tier 3 and allows crediting for projects with a median albedo benefit such that there is an increase in climate mitigation due to changes in albedo. The column "Difference from Current Projected Credits" depicts the impact on credit yield across all projects for each tier deployed.

could use the Hasler data's median albedo deduction layer, which makes assumptions around most likely maximum carbon storage and land use transitions. The Hasler data also generated single-transition albedo change maps that could be matched to project-specific land cover changes (e.g., crop to evergreen broadleaf forest) when they are well defined, which could be employed to replace assumptions around starting and ending conditions and carbon storage. When we use the 100% threshold and deduct the albedo impacts from the eligible projects' credits using the median Hasler data albedo deduction layer, the projected credits across all projects would be reduced by 46.5% (Table 1 and Fig. 4). A discounting of $CO_2$ by albedo change combined with a 50% ineligibility threshold would reduce projected climate mitigation by 49.0% (Table 1 and Fig. 4).

For Tier 4, we propose that projects monitor, verify, and report albedo benefits in addition to deductions, and that these benefits could become valued for their contributions to climate change mitigation. Sixteen projects (9.3%) in our analysis showed the potential for albedo benefits, indicating that their net climate benefit could be greater than expected from $CO_2$-only estimates. These projects could have more than 3.8 million metric tons $CO_2e$ of unaccounted climate benefit due to albedo changes (Table 1 and Fig. 4). If we assume albedo benefits are valued similarly to ARR $CO_2$ removals at $15.60 per credit[28], this translates to more than $59 million in additional income to support project scaling, project longevity, and local communities. In comparison, under the same assumptions for credit price, Tier 3 with a 50% ineligibility threshold (Table 1) would result in more than $6 billion less going to projects in places with albedo deductions. We place this tier of ambition fourth because higher-resolution datasets and deeper protocol development would be needed to justify crediting albedo benefits while erring on the side of more conservative estimates[5,29]

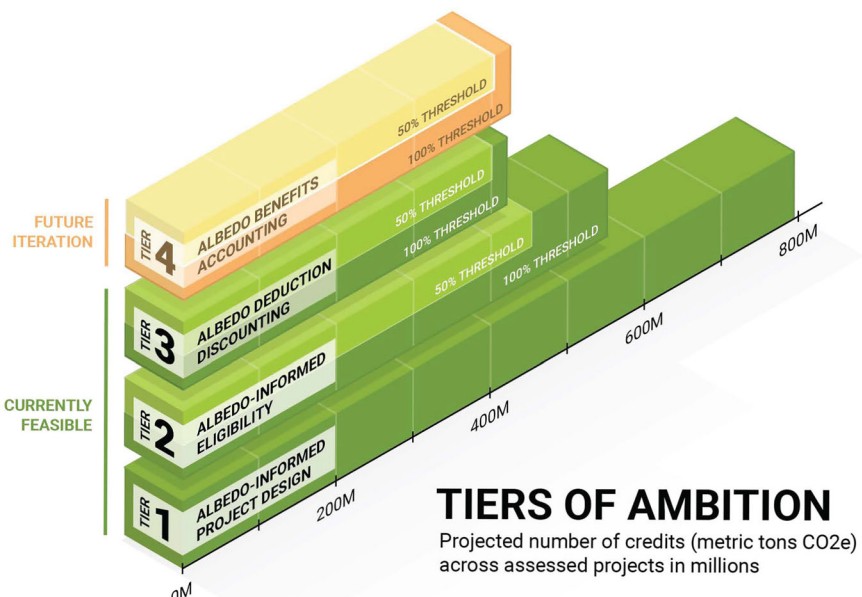

**Fig. 4 | Tiers of ambition for albedo accounting in carbon market protocols, depicting projected credit production across the 172 projects assessed.** Tier 1 incorporates albedo into project planning, and is represented by the total projected credits, in million metric tons $CO_2$, reported by the project descriptions of all 172 projects. Tier 2 introduces a threshold of albedo deduction beyond which projects would be ineligible (an "ineligibility threshold"), and shows two scenarios of ineligibility thresholds: 50% and 100%. Tier 3 builds on Tier 2 and discounts credits by the albedo deduction after removing ineligible projects based on the ineligibility threshold. Tier 4 builds on Tier 3 and allows crediting for albedo benefits. Tiers of ambition for albedo accounting in carbon market protocols © 2024 by Vin Reed is licensed under CC BY 4.0. To view a copy of this license, visit https://creativecommons.org/licenses/by/4.0/.

(Fig. 4). When looking across all projects assessed, the potential albedo benefits found in some did not make up for the albedo deductions found in others (Fig. 4).

## Discussion

Our analysis suggests that 12% of current VCM ARR projects occur in places where albedo entirely negates the mitigation benefit, and a quarter occur in places where albedo halves the mitigation benefit. Yet, the majority are concentrated in places where albedo changes are expected to be minimal, and 9% of projects occur in places where albedo would augment the mitigation benefit. While our analysis has uncertainties, it shows the importance of incorporating albedo in carbon credits from ARR, and we provide a tiered approach for doing so.

One principal source of uncertainty stems from our consideration of only $CO_2e$ and albedo, and not other biophysical factors. Changes to albedo are one of several potential biophysical outcomes of ARR activities that may impact a project's net climate impact[30,31], such as changes to cloud cover, evapotranspiration rates, vegetation moisture storage, surface roughness, and snow cover duration[6,10,32,33]. Not all of these mechanisms influence global climate in a way comparable to the global impacts of $CO_2e$ or albedo changes, with several instead altering only the local surface energy budget. These can involve complex interactions and feedback between surface conditions and the atmosphere, involving temperatures, humidity, soil moisture, and clouds, and with variability linked to ecoclimatic and biome settings[34]. For example, increases in local evapotranspiration-related cooling due to ARR are modest in drier biomes, and the heat can be retained within the regional system when water vapor condenses, negating the cooling effect[20,25,34]. This suggests that evapotranspiration may play only a small role in the climate forcing of ARR in drier areas, which are also some of the areas with the greatest albedo deductions. In wetter, more humid areas, where there is a relatively large transpiration cooling

effect from increased tree cover, the risk of undercounting climate-positive effects from non-albedo biophysical effects is likely largest, though it remains unclear whether ARR would produce clouds that have a net cooling effect on the global climate system[33,35,36].

Overall, there remains uncertainty around the global warming or cooling magnitude of these additional, local biophysical effects of ARR activities[20,36–38] and more work is needed to include them quantitatively within VCM accounting. Furthermore, recent studies suggest that albedo change dominates the biophysical radiative forcing change from ARR activities at a variety of scales[39]. Although incorporating different factors could change the magnitude of our results (and potentially tip projects into or out of eligibility if they are right at the threshold), we expect the general patterns to hold even as science improves. Moreover, because the bulk of the non-carbon and non-albedo effects would augment the mitigation benefit, it is conservative for VCM accounting to not include these effects until uncertainties are more resolved[5]. In recognizing the value of these local biophysical benefits to people and ecosystems[40], we join others in calling for such work to be prioritized by the scientific community[8,25,36]. This practice of VCM protocols providing conservative, default datasets (such as the Hasler data) upon which improvements and project-specific contexts can be proposed—such as deforestation risk maps[41] or leakage accounting defaults[42]—can be a path to near-term implementation while creating opportunities for improved data where viable.

Other sources of uncertainties stem from projected carbon accumulation and assumptions around starting and ending land covers. We used publicly reported carbon credit projections; however, uncertainty around ex ante carbon credit projections are only partially addressed within existing ARR protocols. Upon credit issuance, protocols typically require reporting of sampling error but inconsistently require propagation of other error sources such as measurement error (e.g., measurements of stem diameter at different locations over time) or model error (e.g., selection of inappropriate or uncertain allometric

models)[43,44]. Most projects do not report robust uncertainty around projected (ex-ante) estimates of credit production, so we could not directly quantify uncertainty associated with the projected carbon credits. The ex-ante estimates used here are also likely to be refined before verified credits are issued and will vary depending on the successes of the ARR projects.

In many instances, the carbon data may have greater uncertainty than the albedo data[14,17]. Albedo data benefits from being directly quantifiable from satellite data, whereas carbon sequestration and storage are difficult to estimate accurately[7]. However, there are several sources of uncertainty around the estimates of albedo change used here. The Hasler data was generated according to the following broad steps: first, generating global maps of "mostly likely" land cover transitions; second, comparing the $CO_2e$ of the albedo-driven radiative forcing to the radiative forcing from $CO_2e$ associated with maximum carbon storage possible for a given location[17]; and third, calculating the ratio of albedo- and carbon storage-$CO_2e$ changes (see additional details in the "Methods" of Hasler et al.). Our use of the albedo deduction/benefit layer thus assumes that the "most likely" transition per the Hasler data and the geographic extent per the Karnik et al. data[45] used for project boundaries reflects the actual transition for each VCM project. However, different pre- and post-ARR land covers can alter the albedo change attributable to a project[26], including different ways baseline or historical albedo are defined. Moreover, the 500-m resolution of the Hasler data may fail to capture on-the-ground variation in land cover. Project-specific albedo assessments at finer resolution and with project-specific baseline assumptions would produce more accurate results[26,38]. For example, a project could select the correct pre- and post-land cover transition from the Hasler data single transition layers and use further-refined estimates of projected carbon sequestration.

For the uses of albedo data proposed here, reliability is most critical if albedo accounting changes project eligibility or credit generation. Thus, we view Tier 1 as the most immediately implementable because albedo information is being used to steer projects towards places that are more likely to be climate positive, but does not alter eligibility or carbon credits. Incorporating Tier 1 considerations into project design and assessment is immediately available to and reproducible for project developers, reviewers, and purchasers pursuing the highest quality of ARR credits, regardless of whether protocols adopt formal considerations of albedo effects.

Tier 2 requires the next level of reliability in albedo information, especially for projects that occur along the threshold of ineligibility. In those locations, higher resolution data or more complete accounting for other biophysical factors could shift projects in or out of eligibility. However, Hasler and colleagues (2024) found that accounting for variation in the albedo data did not substantially shift albedo deductions/benefits[14]. They found that variation around different radiative kernels shifted albedo deduction/benefits by no more than ±15% in most places, with greater variability in boreal biomes. For only 9% of global land area did this variability shift pixels over or under a 50% albedo deduction threshold.

Notably, there are reasons why lands with substantive albedo deductions may be preferred. For example, they may be lower cost and/or come with important co-benefits beyond climate mitigation. A strict threshold would exclude these locations, whereas a less strict threshold would allow these locations to remain with a more complete estimate of their climate mitigation potential (this paper's Tier 3). The albedo data available today makes Tier 2 reproducible and operational in the near-term with relatively simple edits to VCM protocols (e.g., adding albedo-based eligibility constraints).

Higher-resolution and/or more project-specific data on albedo effects may be most important for Tiers 3 and 4 because its inclusion would alter the carbon credits for many projects. As we note, reliability can be improved at the project level by using more precise land cover

transitions and carbon estimates. Given the uncertainties noted around warming and cooling effects of other biophysical changes of ARR beyond albedo, these tiers would benefit from additional scientific and data development in coordination with policy iteration within the VCM, the result of which could be a more complete net biophysical accounting of ARR activities. Moreover, albedo benefits do not directly reduce ocean acidification in the way that atmospheric $CO_2$ removal does, highlighting a distinction of climate change mitigation approaches that target holistic atmospheric restoration beyond just climate impacts[46].

All tiers will require collaborative communication and education with market stakeholders who may not be familiar with biophysical factors, since climate mitigation activities historically have focused specifically on $CO_2e$. Additionally, deeper tiers of ambition will benefit from and in some cases require additional research, datasets, tools, and policy to become feasible. Development of an operational Sentinel-2 and Landsat albedo product[47–50] at 20- to 30-m resolution with global, seasonal coverage has the potential to enhance the Hasler data and provide albedo deduction/benefit information at spatial scales more directly aligned with ARR projects.

To be clear, we view these tiers of albedo accounting as progressively ambitious iterations of applied science within carbon markets. Participants in the carbon markets should be held accountable to the best available science, which requires continuous protocol improvement[5]. This rapid iteration is happening, for example, with the uptake of projects using VM0047 and other new protocols that incorporate dynamic baselining after academic studies demonstrated gaps in traditional baselining approaches[51–53]. Similarly, tools that calculate non-permanence risk have been updated in light of scientific discourse on the need to incorporate future climate change impacts[54]. And, while this paper was under review, a new ARR protocol by Isometric was released that follows a similar approach to our Tier 2, with a 100% threshold[55], demonstrating potential for market uptake of albedo accounting.

In this vein, we acknowledge that adoption of higher tiers may be limited at this point, but we include them because they show a fruitful direction for future design and implementation, for which we encourage continued research and policy development. Notably, it appears that implementing Tier 2—making ineligible the portion of projects at greatest risk for especially large albedo deductions—may resolve much of the accounting gap, given the smaller credit reductions as the tiers advance (Table 1 and Fig. 4). Additionally, given that many of the locations with high albedo deductions occur in places where tree cover is inappropriate or ineffective (e.g., afforestation of savanna ecosystems[56] or other arid sites[57]), albedo accounting via Tiers 1 and 2 can help direct planned projects to the most effective areas for ARR. Nevertheless, the holistic accounting unlocked by Tiers 3 and 4 will be important to prioritizing climate finance and accurately valuing NCS. Market labels such as ABACUS[27] could also be used to differentiate projects that meet these tiers, potentially allowing them to access higher prices in the VCM.

Inclusion of albedo accounting within carbon markets will be most effective where protocols implement consistent, reproducible, standardized processes for its inclusion. The Hasler data was generated by an independent third-party, is publicly available and globally consistent, and was created with reproducible and transparent methods. It also comes in a format familiar to many project developers, and thus presents an opportunity for use by all VCM participants, including use cases beyond ARR projects. As with ARR, albedo and other biophysical factors are likely to be relevant to Improved Forest Management, Reduced Deforestation and Degradation, and biochar projects[58,59]. Research for all these cases is needed, including reliable, accurate assessments that are extended to practice with maps and tools. It will be important for these datasets to be dynamically updated, as climate change is likely to continue to influence albedo values, for

example, through reduced snow cover[14,26]. When protocols use the best available science to improve their accounting, market confidence grows and climate finance is likely to reach the projects most effectively mitigating climate change[5,19,60,61].

## Methods

### Searching protocols for current handling of albedo
We reviewed current ARR protocols, available as of June 2024, to identify any existing guidance of handling albedo effects. To do so, we first identified ARR protocols that correspond to all ARR projects on the VCM (see below). We then performed a word search in both the protocols and their parent standard documents; for example, the word search was conducted on both VM0047[53] and the Verified Carbon Standard, v4.7. The terms used in the search were: "albedo," "biophysical," "physical," "biogeophysical," "reflectance," and "radiative forcing." Some instances of "physical" and "biophysical" occurred, but were not in relation to albedo. Otherwise, no mentions or mechanisms for albedo accounting were found in any of the protocols, nor their parent standards.

### Selecting projects for assessment
The VCM ARR projects assessed were all those available as of June 2024 on the open-access database of nature-based carbon offset project boundaries published in Karnik et al.[45]. This database provides a public, open-source, global set of VCM project boundaries, including checks for alignment of project geospatial boundaries against project documentation[45]. From this database, all projects with a "Project Type" of "ARR" (Afforestation, Reforestation, and Revegetation) were selected for a total of 190 projects. Then, 24 that only provided geospatial data in point format—rather than polygons—were filtered out, bringing the total to 166 projects. Upon discussion with the authors of the database, we used the "Project Accounting Area" rather than the "Project Area" when the former was available. Karnik et al. describe the "Project Accounting Area" as the "geographical area of the project that was used to calculate carbon credit issuance," though in most cases, the "Project Accounting Area" and the "Project Area" are the same[45].

Of the 166 projects remaining, 16 projects were dropped from the assessment where they could not produce a quantified median albedo deduction or where ex-ante projected credits could not be found within corresponding project documentation, which was sourced from the carbon registries on which these projects are listed (Supplementary Table 2). This brought the total projects for assessment to 150.

An additional 22 projects were added to those 150 projects to include projects using Verra Methodology 0047[53], which was published in late 2023 and had not yet been added to the Karnik et al. project database. This was done in anticipation of VM0047 becoming increasingly used within the VCM as it replaces other protocols due to its advancement of the widely-recognized dynamic baseline approach to enhancing causal attribution and additionality of a project intervention relative to traditional baseline approaches[51,62]. This brought the total projects assessed to 172. We sourced geospatial boundaries for the VM0047 projects from the Verra Registry. These were converted from KML files to polygon files for the assessment using ArcGIS Pro v3.3.0.

All data was last accessed for this assessment in June 2024 and/or checked in April 2025.

### Albedo deduction data
We obtained albedo deductions and albedo benefits from Hasler et al.[14]. The study provides data on albedo effects in two formats: a "relative" effect, which estimates the albedo-driven radiative forcing as a percentage of maximum sequestered $CO_2e$ from ARR; and an "absolute" effect, which estimates the albedo-driven radiative forcing in units of $CO_2e$. We assessed both data formats and ultimately

determined that while the relative data were appropriate for our global, market-scale analysis, the absolute data were not appropriate for several reasons.

First, there were instances of misalignments in geographic project areas between the Karnik et al. database and the project areas associated with the estimation of projected credits. This is not an issue with the Karnik et al., 2025 dataset (which was extensively vetted[45]), but rather differences in how projects report their boundaries under the VCM registries. For some, the boundaries reflect currently planted lands (while credit projections may also include future plans); others represent areas planned for planting (while credit projections may include only currently planted lands); and yet others might represent broader regions or areas of ownership. Since the absolute albedo effect is determined by the median absolute change (i.e., $CO_2e$ offset) multiplied by the extent of the project boundary, it is critical for the extent of the project boundary to align with the area underlying the projected credits.

The "relative" method (our primary method) is insensitive to misalignments in project boundaries and credit projections under the assumption that the project boundary represents the general type of land (if not the absolute extent) used to calculate the projected credits (e.g., the boundary comprises of mostly pastureland for an afforestation of pastureland project). The relative data are applied directly to the projected credits and thus avoid any issues related to misalignment between the spatial boundary file on the registry and the projected credits in the project documentation. We therefore chose to employ these data in our analysis.

We obtained our relative albedo deductions and albedo benefits from the raster dataset ("AlbedoOffset_005.tif") published by Hasler et al. Positive values in the dataset indicate albedo deductions, whereas negative values indicate albedo benefits. The albedo dataset caps values at ±10,000% to avoid ±infinity[14]. We use the term "albedo deduction" rather than "albedo offset" to avoid confusion with emissions offsets associated with carbon credits.

### Summarizing by biome and protocol
We examined how project-level albedo deductions and benefits interact with biomes and the protocols that projects are registered under. To examine whether albedo deductions and benefits were more likely to occur within a given biome, we overlaid the project boundaries on a global dataset of biomes and summarized the project hectares located in the biome, the expected credits, and the minimum, maximum, and median albedo offset (Supplementary Table 1). Biomes were sourced from the RESOLVE ecoregions dataset[63]. Of the 14 biomes in the dataset, 10 had more than one project in our assessment, and the other 4 biomes were excluded from this analysis. Where projects spanned multiple biomes, we only included the portion of projects' hectares that intersected with each biome (Supplementary Table 1).

For projects that spanned multiple biomes, we attributed projected credits to each biome by assuming the proportion of projects' area in each biome aligns with the proportion of projects' projected credits in that biome. To arrive at this, we calculated the proportion of the project hectares within each biome relative to the total hectares of those projects. For example, if project X has 100 hectares and 75 of those intersected with Biome A, we included 75% of project X's projected credits to Biome A's projected credits, and the other 25% in the other biome(s) project X intersected.

For protocols, we classified each project by the protocol under which carbon credits were generated. We similarly summarized the project extent, expected credits, and median, maximum, and minimum albedo deductions for each project (Supplementary Table 2).

### Projects' ex-ante credit projections
Project documentation for the 172 projects was downloaded from the respective registries to retrieve the reported ex-ante projected credits.

Where multiple project documents were available with ex-ante projected credits reported, the most recent document published was used. Caution should be noted with the use of ex-ante credit projections, as these projections from a project are often refined before credits are issued ex-post (i.e., after the project activity has occurred, appropriate protocols have been used to quantify successful carbon sequestration, and quantification is validated by an accredited third-party). Our assessment was performed using these ex-ante projections rather than ex-post issued credits, as in a majority of cases, projects have not yet issued credits. In most instances, ex-ante estimates of credit production employ locally-parameterized predictive models and are used to garner early-stage project investment, and thus we assume that—although uncertain—they are reasonable proxies of the ex-post credits to be produced.

## Applying the albedo deduction

We used the zonal statistics function to calculate the median albedo deduction/benefit within each project boundary for all ARR projects. Where the median albedo deduction for a project was null, the project was dropped from this analysis. This is a potential source of under-reporting albedo deductions, as areas within the albedo deduction raster dataset with null data are often deserts, which would have high albedo deductions if reforested[14].

The median albedo deduction for each project was applied as a percent deduction on the ex-ante projected credit production sourced from project documentation. For example, where the median albedo deduction for a project's area was 60%, the project's total estimated $CO_2e$ climate benefit was multiplied by 0.4 to derive an albedo-adjusted estimate of that project's total estimated $CO_2e$ climate benefit.

## Reporting summary

Further information on research design is available in the Nature Portfolio Reporting Summary linked to this article.

# Data availability

All input data are publicly available. Project descriptions were sourced from the registries listed in Supplementary Table 2. Project boundaries sourced from Karnik et al. were retrieved from https://zenodo.org/records/11459391. Project boundaries sourced from projects using VM0047 were sourced from. KML files on the Verra Registry at https://registry.verra.org/app/search/VCS. The albedo deduction/benefit raster layer used to calculate project median albedo was sourced from Hasler et al., 2024, at https://dataverse.harvard.edu/api/access/datafile/8550241.

# Code availability

The data and associated R scripts generated in this study have been deposited in the Zenodo database and can be accessed at https://doi.org/10.5281/zenodo.16749322.

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

## Acknowledgements

Thanks to Catherine Chamberlain, Sebastian Busby, Richard Campbell, and Natalia Mushegian for helpful feedback on early drafts of the manuscript. Thank you to the American Forest Foundation for funding the graphic design of Figs. 1 and 4 and for enabling L.M.R. to dedicate time to this research. And thank you to the American Forest Foundation and Oregon State University for funding publication as open access.

## Author contributions

L.M.R. conceived of the idea. L.M.R. and J.J.B. performed the data gathering and analysis. J.J.B. created the graphs and map. J.J.B, S.C.C.-P., L.P.A., C.J.S., and C.A.W. advised throughout the work, and L.M.R., J.J.B., S.C.C.-P., L.P.A., C.J.S., and C.A.W. contributed to the manuscript.

## Competing interests

L.M.R. and S.C.C.-P. belong to organizations that manage VCM projects implementing reforestation. S.C.C.-P. is on a Technical Advisory Board for a VCM coalition. J.J.B. is a science advisor for a company that

consults on the science and technology of $CO_2$ management, including forestry projects. The remaining authors have no competing interests to declare.
