## [Transparent Peer Review file · Nature Communications]

Accounting for Albedo in Carbon Market Protocols

Corresponding Author: Ms Lynn Riley

Version 0:

Reviewer comments:

Reviewer #1

(Remarks to the Author)

COMMENTS TO THE AUTHOR(S):

This paper compares the albedo impacts of 172 Afforestation, Reforestation, and Revegetation projects for assessing natural climate solutions deployed globally for climate change mitigation. This paper suggests continuing work from previously published research also in Nature Communications, yet the manuscript can benefit from a more thorough literature review and better contextualizing of the results. The literature review seems a little inadequate and despite analyzing multiple unique projects globally on how changes in albedo from ARR projects can affect the climate on a global scale, much of their research references their own previous work, specifically two papers. In general, the paper Methodology and Results appear sound (with some caveats outlined below), but would benefit from a bit more work on the Introduction and Discussion and some details added to the Methods and Results. Please see below for detailed comments.

Introduction:

Much of the introduction lacks outlook and further examination of other previous work. Much of the previous research referenced stems from the authors past research, with other research cited being in the past 5 years. The research of effects of forests on the landscapes in relation to surface reflectivity is admittedly not a well-discussed topic, but has been around for some time, and with this paper focusing on the global outlook of ARR and its role of albedo within it, I expected to at least see some reference to other known studies. I have listed some below:

Betts, R. (2000). Offset of the potential carbon sink from boreal forestation by decreases in surface albedo. *Nature*, 408, 187–190.

Kirschbaum, M. U. F., Whitehead, D., Dean, S. M., Beets, P. N., Shepherd, J. D., & Ausseil, A. G. (2011). Implications of albedo changes following afforestation on the benefits of forests as carbon sinks. *Biogeosciences*, 8(12), 3687-3696.

Line 70-75: I find some of the authors reasonings falling flat. The first reason was simply increasing interest in credits for CO₂ removal, yet, nowhere within the paper, other than the definition of carbon crediting, provided no justification as to when, why or where the interests for CO₂ credits has spiked, especially with the applied stigma (i.e., carbon credit trading, legally emitting more GHGs). The second reason stems simply from new datasets being available. As this paper is a continuation of previous work, that is understandable, as the authors make mention in the methods of using rasterized data from a previous paper (line 561). However, I feel the authors can make a better-defined explanation as to why the use of this now available data makes it novel as to be utilized. I believe the authors can do a more effective job at expressing their reasoning and objectives behind this study more clearly.

Figures:

I strongly appreciate your Figure 1 conceptual illustration from the combined process of changes in albedo and carbon sequestration over the course of an ARR event. Your figure effectively articulates how albedo can affect global warming impact on a short term due to the lowering the surface's reflectivity, but depending on the success of the AAR project, will be supplemented with the magnitude of carbon removal over the longer term. This figure will be a great one to reference in the future when stakeholders want to raze forests to plan monoculture corn.

Methods:

Why were albedo deductions beyond 300% excluded? Were there many outliers, which in your box and whisker plot can definitely skew your results? I did not see to find a reasoning for this in your methods, nor supplementary.

Results:

Line 105: Within your Protocol Assessment (Figure 3B), what do you mean by Gold Standard: Afforestation/Reforestation? Simply listing a link within your supplementary doesn't suffice.

Lines 112-196: Your threshold (Table 1) and tiers (Figure 4) seemed out of place, as they were not introduced or explained anywhere else in your manuscript, such as why the authors used these tiers, and any information including advantages or how they are determined.

It was also difficult to comprehend your final heading "Difference from Current Projected Credits". Much of your results read like methodology, but without much clarity. I would recommend revamping your results so they are clearer, and focus on the significant results of the work which are harder to find.

Discussion:

From figures 2 and 4, I was hoping to see a discussion of how we can potentially increase the albedo benefit, either based on biomes, land use change, and/or related to current studies which have had advantageous (or setbacks!) progress towards projected carbon credits when accounting for albedo benefits through land cover transitions. You touched on it briefly (Lines 165-167), but I think this would be a great way to pull both your feasible tiers/thresholds together with currently completed spatial ARR projects observed within your study.

I look forward to your resubmission.

Reviewer #2

(Remarks to the Author)

The manuscript explores the climate impact market beyond carbon of reforestation projects in the voluntary carbon market (VCM). It does so by overlaying two recently published datasets - project boundaries of VCM projects with a layer that maps the radiative impact of albedo changes following land cover change (i.e. grassland to forest). The manuscript then lays out an approach for including albedo change into VCM accounting and puts this not only in a climate-impacts perspective but also lays out the financial implications of such a move away from methodologies entirely focussed carbon accounting. It shows that accounting for radiative forcing impacts beyond that of CO₂ can have far-reaching implications for reforestation projects participating in the VCM.

The manuscript is clearly written and aims to be policy relevant and moves the idea of turning carbon accounting into a more holistic climate accounting, and helpfully actually lays out a potential approach and the implications of that approach.

One major point that I think is missing here (and in the Hasler paper) is to clarify that this addresses non-local albedo effects of land cover change but does not account for regional non-radiative effects (which are much more challenging to convert to CO₂e). At a local-to-regional scale non-radiative processes such as evapotranspiration (ET) and surface roughness (SR, as a mechanism for vertical mixing of air) can counter albedo effects. Especially in the moist tropics where most of these projects are located increase in forest cover leads to local to regional cooling. And while the authors diligently avoid talking about temperature change as this manuscript is about radiative CO₂e impacts, this local effect is important to nature and people living in those areas. I would like to see this distinction between the global and local effects of pointed out in the discussion given the broader policy scope of the manuscript.

I couldn't find any information on the resolution of the Hasler data in the manuscript and the Methodology section is rather short, to make the manuscript more self-contained I would suggest adding some more detail on how the albedo dataset has been generated. Also 500m is fairly low resolution considering the diverse land cover some of those projects are located in. Emphasizing this when discussing the uncertainties would be useful (beyond the point made around using higher resolution data sources in the level 2 assessment).

Minor points

1. Figure quality (might be just in the draft)
2. "Sources of uncertainty" and following text could be moved to Discussion
3. Ln 294 sentence is unclear

Reviewer #3

(Remarks to the Author)

NCOMMS-24-69859-T

Key results

This manuscript addresses for the first time the potential effect of albedo on the net climate impact of reforestation projects enrolled in the voluntary carbon market. The analysis suggests that expected long-term albedo reductions from these reforestation projects could cancel a non-trivial portion of their climate benefit. The authors propose integrating albedo impacts into market rules governing project eligibility as well as potentially deducting albedo impacts from final credit

issuance.

Validity

The analysis seeks to extend the domain of climate accounting for reforestation beyond carbon removal. For 10% of projects, the analysis concludes that albedo impacts fully negate the project carbon removal. As these projects are generally located in arid or grassland biomes with high albedo and low historical tree cover, the case here is quite clear. The analysis, however, also suggests smaller, but still non-trivial albedo impacts, for reforestation projects in tropical and temperate broadleaf forests, implying the projects are over-credited and require albedo deductions.

This conclusion is premature without considering other land-atmosphere interactions, such as the well-documented cooling effects of evapotranspiration (Nobre et al., 1991; Shukla et al., 1990), particularly in broadleaf forests. In broadleaf biomes, it is possible that evapotranspirative cooling substantially or fully offsets the warming from reduced albedo. The authors do mention these potentially compensating effects in their Discussion.

"However, other biophysical factors can substantially impact net climate impacts of NCS projects either augmenting or further diminishing the benefits (Ellison et al, 2024; Zelinka et al, 2020). These include enhancement of cloud cover, local changes in surface cooling from altered evapotranspiration rates, vegetation moisture storage changes, changes in surface roughness with effects on heat transfers, and changes in snow cover duration (Nabuurs et al, 2022; Zickfield et al, 2023; Cerasoli et al, 2021)."

Furthermore, the baseline forest extent underlying the estimates of albedo change is not made clear. For example, the Hasler et al. 2024 map (Fig. 2) shows large albedo deductions for the entire boreal forest region. Replanting cleared boreal forest will obviously drastically lower albedo relative to snow, in other words, if the estimated albedo deduction is computed relative to present-day land cover. However, this baseline condition for estimating albedo change is a policy or value-based decision. As the authors acknowledge, nature-based solutions for climate have the twin objectives of both reducing atmospheric carbon and protecting ecosystems, and are often explicitly linked to initiatives such as 30 by 30 (protecting 30% of terrestrial ecosystems by 2030). Accordingly, the baseline for computing albedo change could also justifiably be defined as the historic albedo prior to widespread land clearing (Hurt et al., 2011, 2020). The manuscript would benefit from more directly stating this baseline albedo assumption. This assumption, for example, effectively nullifies all boreal reforestation.

Significance

The analysis demonstrates convincingly the need to consider albedo in siting reforestation projects. This is a valuable result that will certainly influence the planning and development of reforestation projects. There does seem to be an inordinately large number of projects in fairly arid places (Fig. 2) that historically have not had much tree cover. The aridity alone invites the question of failure risk for these projects and potential overestimation of expected carbon uptake even if successful.

As this manuscript is the first of its kind pertaining to the carbon market, its conclusions will likely elicit calls for integrating some of the authors' recommendations into market rules. The suggested improvements below are aimed at helping non-technical market stakeholders grasp the core assumptions and limitations underlying the analysis, particularly on three main issues:

1. Albedo is just one component of a project's total non-carbon effect on land-atmosphere interactions, and not necessarily the largest. Albedo reductions could be offset by other land-atmosphere interactions, such as increased evapotranspirative cooling.
2. The expected albedo changes assume a baseline of present day non-forest cover as opposed to historical forest cover.
3. It remains unclear whether albedo deductions (and conversion to carbon units for credit deduction) can be calculated operationally with sufficient rigor and reproducibility for an individual reforestation project.

Data & methodology

The manuscript's Supporting Information section Testing the Albedo Radiative Forcing Change Method highlights the potential challenges to rigorous, reproducible quantification of albedo deductions, particularly conversion to carbon units. Although the manuscript's primary analysis uses the Hasler et al. 2024 albedo reduction dataset in relative (%) units, the authors also test the Hasler et al. dataset in absolute units of change in radiative forcing change (CO₂e). This analysis using absolute CO₂e units produces very large, nonsensical numbers (e.g., -35 million tons) that would cancel the carbon removal for all reforestation projects by at least an order of magnitude. A more thorough investigation of this discrepancy is critical.

Analytical approach

The analysis proposes extending the accounting domain for reforestation projects beyond carbon sequestration to one of many possible land-atmosphere interactions, in this case albedo. The analysis reports summary statistics of albedo deductions by biome and by carbon registry.

However, as mentioned above, other land-atm interactions, not included in the analysis, such as evapotranspiration, could

be sufficiently large to offset the manuscript's estimated albedo impacts, potentially rendering moot the need for albedo-specific accounting.

Moreover, the manuscript would benefit from some quantitative evaluation of uncertainty in the Hasler dataset, although I understand that is likely beyond the scope of this analysis.

Suggested improvements

1. I suggest a deeper investigation of the enormous discrepancy in results between the two Hasler et al. datasets: relative albedo deduction versus absolute change in radiative forcing. If it's not possible to resolve this discrepancy within the scope of this manuscript, I suggest at least clearly diagnosing the reason for it and reporting it in the main text.

2. Relatedly, I suggest a bit more description of the data and methods underlying the Hasler dataset (i.e., specific calculations, assumptions, limitations). Some description of the sources and magnitudes of uncertainty in the Hasler dataset would help the reader better evaluate this manuscript's results.

3. I suggest more explicitly enumerating the analysis's main assumptions/priors (mentioned above):

i. The analysis does not present a complete accounting of project non-carbon impacts. Unaccounted climate impacts could offset the reported estimates.

ii. The analysis assumes present vs. historical tree cover for computing albedo change.

4. In light of the above, and the difficulty of computing consistent, rigorous albedo deductions (as evidenced by the discrepant results between the Supporting Info and main text), I suggest a more nuanced discussion of the actual need for albedo deductions (beyond a project eligibility threshold: proposed Tiers 1 & 2) and a realistic appraisal of the challenges to reproducible implementation of albedo deductions.

Clarity and context

The manuscript's focus on albedo is understandable, as a complete accounting of climate impacts would incorporate numerous land-atm interactions too complex/uncertain to quantify for a single reforestation site, let alone operationalize in a consistent, reproducible fashion for carbon crediting. The authors use an oft-repeated refrain that is, in fact, at odds with the reality of the carbon market.

"Participants in the carbon markets should be held accountable to the best available science without invalidating previous efforts that used the best science available at that time."

Buyers enter the market seeking to purchase real emissions reductions. An appeal to the "best available science" of the past does not rectify the loss of millions of dollars on credits rendered worthless. Buyers simply stop purchasing credits.

So the question becomes: what accounting domain is sufficiently large to ensure robust, conservative crediting in the aggregate, but sufficiently well-defined to be tractable. Tractable in the following sense:

1. All crediting calculations (or algorithms) can be validated against independent measurements to ensure consistent, reproducible crediting.

2. Uniform, conservative values are assumed for any crediting quantities for which consistent, reproducible calculation is infeasible.

I suggest the Discussion section evaluate the manuscript's proposed albedo deductions to crediting within this framework, or at least acknowledge more directly and grapple with these very real market constraints/realities.

References

Yes, the manuscript contains appropriate references.

Your expertise

Please indicate any particular part of the manuscript, data or analyses that you feel is outside the scope of your expertise, or that you were unable to assess fully.

The manuscript uses Hasler et al. 2024 for quantification of expected albedo change due to reforestation (Hasler et al., 2024). A detailed assessment of the quality/methods underlying this dataset is beyond my expertise.

REFERENCES

Hasler, N., Williams, C. A., Denney, V. C., Ellis, P. W., Shrestha, S., Terasaki Hart, D. E., Wolff, N. H., Yeo, S., Crowther, T.

W., Werden, L. K., & Cook-Patton, S. C. (2024). Accounting for albedo change to identify climate-positive tree cover restoration. *Nature Communications*, 15(1), 2275.

Hurt, G. C., Chini, L. P., Froking, S., Betts, R. A., Feddema, J., Fischer, G., Fisk, J. P., Hibbard, K., Houghton, R. A., Janetos, A., Jones, C. D., Kindermann, G., Kinoshita, T., Klein Goldewijk, K., Riahi, K., Shevliakova, E., Smith, S., Stehfest, E., Thomson, A., ... Wang, Y. P. (2011). Harmonization of land-use scenarios for the period 1500–2100: 600 years of global gridded annual land-use transitions, wood harvest, and resulting secondary lands. *Climatic Change*, 109(1-2), 117–161.

Hurt, G. C., Chini, L., Sahajpal, R., Froking, S., Bodirsky, B. L., Calvin, K., Doelman, J. C., Fisk, J., Fujimori, S., Klein Goldewijk, K., Hasegawa, T., Havlik, P., Heinemann, A., Humpenöder, F., Jungclaus, J., Kaplan, J. O., Kennedy, J., Krisztin, T., Lawrence, D., ... Zhang, X. (2020). Harmonization of global land use change and management for the period 850–2100 (LUH2) for CMIP6. *Geoscientific Model Development*, 13(11), 5425–5464.

Nobre, C. A., Sellers, P. J., & Shukla, J. (1991). Amazonian deforestation and regional climate change. *Journal of Climate*, 4(10), 957–988.

Shukla, J., Nobre, C., & Sellers, P. (1990). Amazon deforestation and climate change. *Science (New York, N.Y.)*, 247(4948), 1322–1325.

Version 1:

Reviewer comments:

Reviewer #1

(Remarks to the Author)

Title: Accounting for Albedo in Carbon Market Protocols

COMMENTS TO THE AUTHOR(S):

The authors have spent a significant amount of time revising their manuscript titled “Accounting for Albedo in Carbon Market Protocols”. The authors have addressed certain aspects of feedback, which include much more defined headings and subsections, as well as cleaner transitions and a more streamlined Result and Discussion sections. The Discussion section underlying uncertainties within the paper provides a much clearer picture of the advantages and feasibility of application of different ARR projects. Additionally, the author has made a great effort to be transparent in explaining removal of outliers, explanation of important concepts and provided a stronger foundation for their tiered approaches for current and projected albedo credits.

However, I was disappointed to see that there were other aspects of the manuscript which were not adequately updated. I believe the current literature review is still not substantial. Instead of strengthening the current introduction with the inclusion of new information, citations and references were only instead added to current literature. Many of the sections of the manuscript which were highlighted as ‘updates/new text/explanations’ were already present before the initial review (and confirmed when revisiting the first iteration of the submitted manuscript), with very little to no improvements in context or revision.

The authors did add a few additional sentences which discussed their focus on ARR within the Introduction, but greater care and detail is needed for expanding on exactly what are the inadequacies of current research. and where in the literature this manuscript and work is filling a knowledge gap.

I believe the manuscript still warrants a strong merit for publication with these feedbacks more satisfactorily addressed.

Other:

Line 75-76: Datasets that quantify albedo effects from ARR are now available, and albedo change from ARR is expected to be consequential how? Simply listing them as a reason without a proper explanation is more sufficient. Elaborate in greater detail why ARR data is becoming more available, why is this important, and why it is becoming more mainstream over other types of nature-based solutions.

Line 84: change ‘are’ to ‘were’.

Line 226: What other biophysical factors are you referring to?

Line 470-471: Rephrase this sentence to better understand your project proportions.

Reviewer #2

(Remarks to the Author)

I have no further comments and thank the authors for their thorough re-working of their manuscript.

Reviewer #3

(Remarks to the Author)

The authors’ have reasonably incorporated the suggestions/feedback provided by the reviewers in the form of revised text.

The main analysis – extraction of a project-level albedo deduction from the Hasler et al. 2024 dataset – is straightforward.

Policy recommendations, of course, are intrinsically more subjective, and therefore need not hold up publication. The authors' Tier 1 and Tier 2 proposals – effectively an optional or mandated regional filter on siting reforestation projects to avoid large albedo reductions – is eminently sensible. Such a filter could even be achieved using a simple map of historical forest cover, particularly to avoid projects in grassland and arid biomes with little chance of durable carbon storage, possibly negative biodiversity impacts, and large albedo effects.

On the question of computing albedo deduction relative to historical or present-day albedo (most relevant for boreal reforestation), the authors simply offer the justification of conventional practice. The implication is that a forest carbon credit should be a contractual obligation encompassing not only carbon impact, but total radiative forcing impact.

A discussion of the minimum required properties of an asset for sustaining a functioning, liquid asset market is obviously beyond the scope of the manuscript. However, any asset with sufficient confidence for sustaining a market requires defining an accounting boundary within which all underlying asset calculations are standardized, reproducible, and independently verifiable (the data not simply the calculations must be verifiable). In other words, this accounting boundary defines what is computationally possible.

The authors' Tiers 3 & 4 recommendations run up against these limits. Although they recur to the phrasing that future science will improve, quantification and attribution of more atmospheric processes to individual projects (i.e., extending the asset accounting boundary to encompass radiative forcing) approaches the fundamentally unresolvable/unquantifiable. The gap lies not in the state of the science, but in the disconnect between the scale of an individual reforestation project (often on the order of a few hundred to thousand hectares) and the scale of atmospheric/radiative processes.

The forest carbon market is at present trivially small, and will never scale without a more strict definition (and calculation) of the carbon asset, if the asset remains a carbon offset meant to compensate 1:1 for buyer emissions. A paradigm shift that dispenses with the notion of an offset and transitions to a contribution or fee paid to reforestation would enable more flexible accounting of aggregate climate impact across the entire market (as opposed to individual projects).

This policy critique doesn't bear directly on the manuscript's main quantitative analysis – extracting project-level Hasler et al. albedo deductions for a large sample of existing reforestation projects. I don't wish to delay publication; I simply offer these remarks, such that they are available to the public in the supplementary "peer review file."

Accounting for Albedo in Carbon Market Protocols
Nature Communications submission (NCOMMS-24-69859-T)
May 2025 Responses to Reviewer Comments

Reviewer comments received Feb. 27, 2025 and responses

Reviewer 1

COMMENTS TO THE AUTHOR(S):

This paper compares the albedo impacts of 172 Afforestation, Reforestation, and Revegetation projects for assessing natural climate solutions deployed globally for climate change mitigation. This paper suggests continuing work from previously published research also in Nature Communications, yet the manuscript can benefit from a more thorough literature review and better contextualizing of the results. The literature review seems a little inadequate and despite analyzing multiple unique projects globally on how changes in albedo from ARR projects can affect the climate on a global scale, much of their research references their own previous work, specifically two papers. In general, the paper Methodology and Results appear sound (with some caveats outlined below), but would benefit from a bit more work on the Introduction and Discussion and some details added to the Methods and Results. Please see below for detailed comments.

Introduction:

Much of the introduction lacks outlook and further examination of other previous work. Much of the previous research referenced stems from the authors past research, with other research cited being in the past 5 years. The research of effects of forests on the landscapes in relation to surface reflectivity is admittedly not a well-discussed topic, but has been around for some time, and with this paper focusing on the global outlook of ARR and its role of albedo within it, I expected to at least see some reference to other known studies. I have listed some below:

Betts, R. (2000). Offset of the potential carbon sink from boreal forestation by decreases in surface albedo. *Nature*, 408, 187–190.

Kirschbaum, M. U. F., Whitehead, D., Dean, S. M., Beets, P. N., Shepherd, J. D., & Ausseil, A. G. (2011). Implications of albedo changes following afforestation on the benefits of forests as carbon sinks. *Biogeosciences*, 8(12), 3687-3696.

Thank you for flagging this area of improvement. We agree that additional consideration and examination of previous work would make the paper and introduction stronger. To this end, we have added both papers you recommended, and performed a forward citation search on Betts 2000 (as a key foundational paper) to identify additional relevant materials we may have missed previously. In doing so, we have added 25 references throughout the paper, paying particular attention to the Introduction.

Line 70-75: I find some of the authors reasonings falling flat. The first reason was simply increasing interest in credits for CO₂ removal, yet, nowhere within the paper, other than the definition of carbon crediting, provided no justification as to when, why or where the interests for CO₂ credits has

spiked, especially with the applied stigma (i.e., carbon credit trading, legally emitting more GHGs). The second reason stems simply from new datasets being available. As this paper is a continuation of previous work, that is understandable, as the authors make mention in the methods of using rasterized data from a previous paper (line 561). However, I feel the authors can make a better-defined explanation as to why the use of this now available data makes it novel as to be utilized. I believe the authors can do a more effective job at expressing their reasoning and objectives behind this study more clearly.

Thank you for this note for improvement. We have strengthened our reasoning here by pointing to recent market calls for CO₂ removal credits and expanding ARR activities, and adding a fourth justification that points to prior calls throughout literature for the need for albedo accounting to bridge from science into policy and implementation [Lines 62 – 65; 71 – 80].

Figures:

I strongly appreciate your Figure 1 conceptual illustration from the combined process of changes in albedo and carbon sequestration over the course of an ARR event. Your figure effectively articulates how albedo can affect global warming impact on a short term due to the lowering the surface's reflectivity, but depending on the success of the AAR project, will be supplemented with the magnitude of carbon removal over the longer term. This figure will be a great one to reference in the future when stakeholders want to raze forests to plan monoculture corn.

Thank you. We are happy to hear that the figure will be useful!

Methods:

Why were albedo deductions beyond 300% excluded? Were there many outliers, which in your box and whisker plot can definitely skew your results? I did not see to find a reasoning for this in your methods, nor supplementary.

Thank you for catching this. We excluded albedo deductions beyond 300% from Figure 3 only (not from the other analysis presented in the paper) for readability of that figure. This removed 6% of projects from Figure 3. We have clarified this in the Figure 3 caption [Lines 115 - 116].

Results:

Line 105: Within your Protocol Assessment (Figure 3B), what do you mean by Gold Standard: Afforestation/Reforestation? Simply listing a link within your supplementary doesn't suffice.

We have added clarity to the caption of Figure 3 to explain what the protocol names (e.g., “Gold Standard: Afforestation/Reforestation”, etc.) refer to, as well as make readers aware that they can access these protocols through links in the Supplementary Information [Lines 113 – 115 and Table S2]. Please note that Gold Standard: Afforestation/Reforestation (GS 2024) is the name of a protocol (i.e., methodology) for producing verified carbon credits from afforestation/reforestation, and protocols often exist under “standards” (i.e., the Gold Standard (GS 2025)).

Lines 112-196: Your threshold (Table 1) and tiers (Figure 4) seemed out of place, as they were not introduced or explained anywhere else in your manuscript, such as why the authors used these tiers, and any information including advantages or how they are determined.

It was also difficult to comprehend your final heading “Difference from Current Projected Credits”. Much of your results read like methodology, but without much clarity. I would recommend revamping your results so they are clearer, and focus on the significant results of the work which are harder to find.

We have revamped our results to better justify our tiered approach, clarify that we developed it ourselves, and describe why we utilize it, which is based on calls for iterative and incremental continuous improvement within VCM protocols (Cabiyo and Field, 2025) [Lines 137 - 147]. We also clarify what “Difference from Current Projected Credits” refers to in Table 1 [Lines 156 - 157]. To summarize, we recognize that immediate adoption of full albedo accounting within existing protocols for producing carbon credits from afforestation may not be feasible. Thus, we presented the tiers as a way to progressively incorporate albedo and iteratively strive for continuous improvement in carbon crediting protocols as has been recently recommended by others (e.g., Cabiyo and Field 2025).

We additionally clarify that the 50% and 100% thresholds we depict are illustrative rather than strict thresholds, and we use them to demonstrate the impact of these potential protocol choices while other thresholds are possible [Lines 170 – 171].

We also have added additional references to Table 1 and Figure 4; clarified our numerical results throughout, and added new headings to our Results to make key findings easier to find, including sections for albedo deductions findings for projects [Line 90], albedo benefits [Line 122], and the effect of albedo deductions on credits [Line 132].

Discussion:

From figures 2 and 4, I was hoping to see a discussion of how we can potentially increase the albedo benefit, either based on biomes, land use change, and/or related to current studies which have had advantageous (or setbacks!) progress towards projected carbon credits when accounting for albedo benefits through land cover transitions. You touched on it briefly (Lines 165-167), but I think this would be a great way to pull both your feasible tiers/thresholds together with currently completed spatial ARR projects observed within your study.

We have added a new section to call out patterns of albedo benefits within the projects we assessed. This added an interesting finding that albedo benefits were more concentrated within a few biomes than albedo deductions, which were more spread out across biomes, pointing to a clear opportunity we were able to call out for ARR activities within those biomes. Thank you for spurring this additional look into albedo benefits. [Lines 122 – 130]

I look forward to your resubmission.

Thank you. We are grateful for your thoughtful comments, which we believe have made the paper stronger, and look forward to any additional feedback.

Reviewer 2

The manuscript explores the climate impact market beyond carbon of reforestation projects in the voluntary carbon market (VCM). It does so by overlaying two recently published datasets - project boundaries of VCM projects with a layer that maps the radiative impact of albedo changes following land cover change (i.e. grassland to forest). The manuscript then lays out an approach for including albedo change into VCM accounting and puts this not only in a climate-impacts perspective but also lays out the financial implications of such a move away from methodologies entirely focussed carbon accounting. It shows that accounting for radiative forcing impacts beyond that of CO₂ can have far-reaching implications for reforestation projects participating in the VCM.

The manuscript is clearly written and aims to be policy relevant and moves the idea of turning carbon accounting into a more holistic climate accounting, and helpfully actually lays out a potential approach and the implications of that approach.

One major point that I think is missing here (and in the Hasler paper) is to clarify that this addresses non-local albedo effects of land cover change but does not account for regional non-radiative effects (which are much more challenging to convert to CO₂e). At a local-to-regional scale non-radiative processes such as evapotranspiration (ET) and surface roughness (SR, as a mechanism for vertical mixing of air) can counter albedo effects. Especially in the moist tropics where most of these projects are located increase in forest cover leads to local to regional cooling. And while the authors diligently avoid talking about temperature change as this manuscript is about radiative CO₂e impacts, this local effect is important to nature and people living in those areas. I would like to see this distinction between the global and local effects of pointed out in the discussion given the broader policy scope of the manuscript.

Thank you for this comment. We agree that clarifying that we focus on non-local albedo effects amongst several biophysical effects that occur at both local and global scales due to afforestation/reforestation is important, and have now stated this more clearly We have added a broader section based on this feedback that explores further non-albedo biophysical effects and their remaining complexity/ uncertainty, calling for additional work by the scientific community to address this so that it could be considered and better valued in project accounting. In the meantime, we offer the following justifications for why our call for albedo accounting is timely: (1) studies show that albedo dominates the other biophysical effects, especially at global scales; (2) it will lead to more conservative carbon project accounting, which the market has been calling for; and (3) that it is feasible to begin accounting for today [Lines 226 – 244]. In the revised manuscript, we reiterate the importance of these local and regional biophysical benefits to people and ecosystems [Lines 256 - 258] and call for continued research in this area [Lines 246 – 262].

I couldn't find any information on the resolution of the Hasler data in the manuscript and the Methodology section is rather short, to make the manuscript more self-contained I would suggest adding some more detail on how the albedo dataset has been generated. Also 500m is fairly low resolution considering the diverse land cover some of those projects are located in. Emphasizing this when discussing the uncertainties would be useful (beyond the point made around using higher resolution data sources in the level 2 assessment).

Thank you for flagging this oversight regarding clearly reporting the Hasler data resolution. We have now clarified in the main text the resolution of the Hasler data (500m) [Line 69].

Additionally, we have added a broad description of how the Hasler data was generated [Lines 280 - 284], and point readers to the Hasler paper for more detailed methods behind that data.

Lastly, we more clearly state that higher-resolution data would generate more accurate results given the geographic diversity within a 500-meter resolution dataset [Lines 287 - 293].

Minor points

1. Figure quality (might be just in the draft)

Thank you for flagging this. We have included images at higher resolution in the updated manuscript, and plan to work with *Nature Communications* to ensure our images meet their resolution criteria (300 dpi). [Lines 42, 104, 112, 209]

2. "Sources of uncertainty" and following text could be moved to Discussion

Thank you for this recommendation; we agree this section is better suited to our Discussion than our Results, where it was previously. We have moved it to the Discussion. We note that *Nature Communications* does not facilitate subheadings within the Discussion section, and so instead we have removed that subheading and used transitional sentences to situate it within the Discussion. [Lines 226 – 331]

3. Ln 294 sentence is unclear

We have made a few edits to this line [now lines 227 - 231] to improve clarity, and elaborate on this point (the importance of biophysical factors beyond albedo) overall [Lines 226 – 258].

Reviewer 3

Key results

This manuscript addresses for the first time the potential effect of albedo on the net climate impact of reforestation projects enrolled in the voluntary carbon market. The analysis suggests that expected long-term albedo reductions from these reforestation projects could cancel a non-trivial portion of their climate benefit. The authors propose integrating albedo impacts into market rules governing project eligibility as well as potentially deducting albedo impacts from final credit issuance.

Validity

The analysis seeks to extend the domain of climate accounting for reforestation beyond carbon removal. For 10% of projects, the analysis concludes that albedo impacts fully negate the project carbon removal. As these projects are generally located in arid or grassland biomes with high albedo and low historical tree cover, the case here is quite clear. The analysis, however, also suggests smaller, but still non-trivial albedo impacts, for reforestation projects in tropical and temperate broadleaf forests, implying the projects are over-credited and require albedo deductions.

This conclusion is premature without considering other land-atmosphere interactions, such as the well-documented cooling effects of evapotranspiration (Nobre et al., 1991; Shukla et al., 1990), particularly in broadleaf forests. In broadleaf biomes, it is possible that evapotranspirative cooling substantially or fully offsets the warming from reduced albedo. The authors do mention these potentially compensating effects in their Discussion.

"However, other biophysical factors can substantially impact net climate impacts of NCS projects either augmenting or further diminishing the benefits (Ellison et al, 2024; Zelinka et al, 2020). These include enhancement of cloud cover, local changes in surface cooling from altered evapotranspiration rates, vegetation moisture storage changes, changes in surface roughness with effects on heat transfers, and changes in snow cover duration (Nabuurs et al, 2022; Zickfield et al, 2023; Cerasoli et al, 2021)."

Thank you for highlighting these potential compensating effects. We agree that they call for more full discussion within our paper. We revisited the literature based on your feedback, and have added the following to better articulate the breadth of non-carbon impacts beyond albedo, some of which could contribute climate mitigation benefits.

First, we better describe non-carbon and non-albedo effects that could affect the net climate mitigation result of an ARR project [Lines 227 - 244], noting conclusions and uncertainties from additional literature.

We also acknowledge the expected directional effect of the addition of these non-albedo and non-carbon considerations, as well as biome and forest type nuances. Given the complexity recognized in the literature, including questions of local vs. global impacts, quantitatively including non-carbon and non-albedo effects remains outside of this study's scope and likely outside the current scope of the VCM to include in accounting today. More work is needed to quantify these additional effects. Because the bulk of the non-carbon and non-albedo effects are directionally similar to the carbon effect (that is, may contribute toward net cooling more often than contributing toward net warming), it is more conservative from a market lens to not include these other effects, at least until tools enabling robust inclusion are available. This mirrors recent arguments in the literature for avoiding projects and protocols that produce over-credited projects (Cabiyo and Field 2025). We add text to make these points and to call for work by the scientific community that addresses these needs, while still calling for the albedo effect to be incorporated in the meantime [Lines 246 - 258].

Additionally, given that it is important for non-technical audiences to understand these limitations, we have also added text to our abstract clarifying that our albedo deductions are non-exhaustive of all potential biophysical effects of ARR activities [Lines 13 – 15].

Thank you again for this comment, which we feel has aided in the transparency of our reporting and in our communication about the limitations of existing data and need for more scientific work.

Furthermore, the baseline forest extent underlying the estimates of albedo change is not made clear. For example, the Hasler et al. 2024 map (Fig. 2) shows large albedo deductions for the entire boreal forest region. Replanting cleared boreal forest will obviously drastically lower albedo relative to snow, in other words, if the estimated albedo deduction is computed relative to present-day land cover. However, this baseline condition for estimating albedo change is a policy or value-based decision. As the authors acknowledge, nature-based solutions for climate have the twin objectives of both reducing atmospheric carbon and protecting ecosystems, and are often explicitly linked to initiatives such as 30 by 30 (protecting 30% of terrestrial ecosystems by 2030). Accordingly, the baseline for computing albedo change could also justifiably be defined as the historic albedo prior to widespread land clearing (Hurtt et al., 2011, 2020). The manuscript would benefit from more directly stating this baseline albedo assumption. This assumption, for example, effectively nullifies all boreal reforestation.

We have clarified the key assumption you point out underlying the Hasler et al, 2024 data in regards to baseline land cover (and generally added more description of some key elements of the Hasler et al, 2024 dataset) [Lines 280 – 286]. We have noted this among the uncertainties that need to be considered given that these baseline assumptions could differ from what is most representative for an actual project. We also now propose that individual projects could carry out more granular analyses by using additional Hasler et al datasets corresponding to single pre- and post-land cover transitions (e.g., from croplands to forests) that match their project activities or considering historic albedo [Lines 287 – 295]. We clarify that while additional steps could be taken for additional projects, our work reports a market-wide analysis and thus requires some alignment across projects. [Line 292]

We acknowledge the reviewer's good point about albedo values prior to land clearance. However, that would not align with how baselines are established within the VCM and/or climate mitigation space. In general, additional climate mitigation is evaluated relative to conditions today and/or recent history (Ellis et al, 2024). We have clarified these underlying baseline assumptions and ways in which a project could conduct a more granular analysis based on their specific project baselines.

The reviewer also makes a good point that there is still value to reforestation in places that may offer limited climate mitigation (for biodiversity and other ecosystem services) which we now mention in the discussion [Lines 256 – 257; 314 - 316]. Further, one of the interesting points in the

Hasler et al. 2024 was that there is climate-positive opportunity within the boreal, demonstrating the value of a more spatially-explicit analysis compared to past efforts where the boreal was uniformly excluded (e.g., Griscom et al. 2017).

Significance

The analysis demonstrates convincingly the need to consider albedo in siting reforestation projects. This is a valuable result that will certainly influence the planning and development of reforestation projects. There does seem to be an inordinately large number of projects in fairly arid places (Fig. 2) that historically have not had much tree cover. The aridity alone invites the question of failure risk for these projects and potential overestimation of expected carbon uptake even if successful.

Agreed. We were also surprised by the concentration of projects in arid locations! With regard to projects falling in arid places, we note that the albedo accounting we suggest can help direct projects to locations that are more appropriate for ARR activities because low-albedo-deduction locations also tend to be more appropriate for ARR efforts (e.g., helps avoid afforestation of savanna ecosystems, for example). [Lines 358 – 361]

As this manuscript is the first of its kind pertaining to the carbon market, its conclusions will likely elicit calls for integrating some of the authors' recommendations into market rules. The suggested improvements below are aimed at helping non-technical market stakeholders grasp the core assumptions and limitations underlying the analysis, particularly on three main issues:

1. Albedo is just one component of a project's total non-carbon effect on land-atmosphere interactions, and not necessarily the largest. Albedo reductions could be offset by other land-atmosphere interactions, such as increased evapotranspirative cooling.
2. The expected albedo changes assume a baseline of present day non-forest cover as opposed to historical forest cover.
3. It remains unclear whether albedo deductions (and conversion to carbon units for credit deduction) can be calculated operationally with sufficient rigor and reproducibility for an individual reforestation project.

We agree with all suggested improvements and thank you for the helpful feedback. With regard to other non-carbon effects, we have added text detailing the potential of these effects to contribute additional climate mitigation, though we note that operational datasets quantifying these effects are not available and large uncertainties remain. We call for continued research on these topics, while still calling for introduction of albedo into VCM accounting in the meantime [Lines 226 - 262]. Given the importance of this point to potential lay readers, we have also added this to our abstract. [Lines 13 - 15]

With regard to baseline assumptions, we have clarified these underlying baseline assumptions and ways in which a project could conduct a more granular analysis based on their specific project baseline (as mentioned above).

With regard to how operational, rigorous, and reproducible our approach is for individual projects, we have clarified that our albedo accounting tiers framework is intended to facilitate a phased transition for albedo accounting, while our main finding points to the market-wide need [Lines 141 – 147; 342 – 343; 366 - 369].

- **Tier 1 (albedo-informed project design) could be implemented today by individual projects without edits to protocols (e.g., it is immediately operational and reproducible given the common Hasler et al, 2024 data, though with lower rigor given the uncertainties we describe in the paper) [Line 296 - 302]**

- Tier 2 (albedo-informed eligibility) could be implemented in the near-term with relatively simple eligibility cutoffs added to protocols based on the reproducible Hasler et al, 2024 data assessment for an individual project (e.g., near-term operational/reproducible given the common Hasler et al, 2024 data, improved rigor beyond Tier 1) [Lines 304 - 319]
- Tiers 3 and 4 (albedo deduction discounting and albedo benefits accounting) would benefit from additional scientific iteration, given the larger repercussions on projects' viability; it would also likely result in the greatest rigor. [Lines 321 - 330; 352 - 354]

Data & methodology

The manuscript's Supporting Information section Testing the Albedo Radiative Forcing Change Method highlights the potential challenges to rigorous, reproducible quantification of albedo deductions, particularly conversion to carbon units. Although the manuscript's primary analysis uses the Hasler et al. 2024 albedo reduction dataset in relative (%) units, the authors also test the Hasler et al. dataset in absolute units of change in radiative forcing change (CO₂e). This analysis using absolute CO₂e units produces very large, nonsensical numbers (e.g., -35 million tons) that would cancel the carbon removal for all reforestation projects by at least an order of magnitude. A more thorough investigation of this discrepancy is critical.

Thank you for this very helpful prompt to revisit the “absolute change” method highlight in our SI. Based on our further analyses, we now believe the absolute change method is inappropriate for our analysis for several reasons. The main reason is that use of the absolute units of change is highly sensitive to mismatches between the area of the shapefile delineating project boundary on the registry and the expected total extent of the reforestation project. The two approaches for calculating albedo changes are:

Absolute albedo change (CO₂e) = project extent (ha) * absolute albedo deduction (CO₂e/ha)

- *Here, a mismatch between the project extent of the project shapefile and the project documentation will produce nonsensical results.*

Relative albedo change (CO₂e) = expected project credits (CO₂e) * relative albedo deduction (%)

- *Here, the estimate is insensitive to mismatches between the area of the project shapefile and the project documentation.*

The discrepancies in project extents are due to misalignment in geographic project areas (reported in a shapefile) between the Karnik et al. 2025 database and the project areas reported in the project documentation, which are used to estimate the projected credits. This is not an issue with the Karnik et al dataset (which was extensively vetted), but rather a reflection of the fact that projects will upload shapefiles delineating varying states of their project to public registries. For example, many projects upload shapefiles that reflect the currently planted area (not necessarily the total area planned for planting), whereas the total area planned for reforestation is used to estimate projected credits in the project documentation.

The “relative change” method (our primary method) is insensitive to these misalignments under the reasonable assumption that the shapefile represents the general land cover (e.g., the boundary comprises of mostly pasture for an afforestation of pasture project). Under this assumption, adjusting the projected credits (which account for planned extents of reforestation) based on a relative albedo change will be valid. We have more clearly stated that this is an assumption underlying our analysis [284 – 287].

Overall, these additional details indicate to us that the absolute method is ill-suited for our market-wide analysis. We have therefore removed results using the absolute method from our SI and moved our description of the approach and our justification for using the relative data in our

Methods [Lines 757 - 793]. Of course, with more accurate details on the project area, type of reforestation, and calculations of credits, the absolute albedo deduction layers for single land-cover transitions (e.g., Hasler et al data for conversion of pastureland to forestland) could provide more sophisticated estimates of albedo effects. However, doing this for our analysis of all projects on the VCM is out of scope as this would require extensive document review and access to modeling details about ARR projects which are unlikely to be publicly available. The relative method is an appropriate alternative that protects against these issues, and we note the opportunity for more accurate, project-specific assessments in [Lines 290 - 295].

Again, thank you for this very important comment, which we feel has substantially improved our results and our messaging around how these data layers might be used.

Analytical approach

The analysis proposes extending the accounting domain for reforestation projects beyond carbon sequestration to one of many possible land-atmosphere interactions, in this case albedo. The analysis reports summary statistics of albedo deductions by biome and by carbon registry.

However, as mentioned above, other land-atm interactions, not included in the analysis, such as evapotranspiration, could be sufficiently large to offset the manuscript's estimated albedo impacts, potentially rendering moot the need for albedo-specific accounting.

Thank you for this comment. Please see our responses above to your first comment under the section "Validity."

Moreover, the manuscript would benefit from some quantitative evaluation of uncertainty in the Hasler dataset, although I understand that is likely beyond the scope of this analysis.

We agree with the reviewer's assessment that a robust quantitative evaluation of uncertainty is beyond the scope of this analysis. However, we have added additional descriptions of how the Hasler dataset was generated, as well as more on sources of uncertainty within the use of that dataset for the use cases we propose, such as if the "most likely" land covers modeled by Hasler et al, 2024, differ from project realities [Lines 279 – 292].

Additionally, we have added magnitudes of variability calculated by Hasler et al, 2024 that could affect our use cases, namely where different radiative kernels from different climate models cause the albedo deductions to change in a way that would affect accounting within our tiers (e.g., would push a project in or out of a 50% eligibility threshold, for example) [Lines 306 - 311]. We also point readers to the Hasler et al, 2024 methods for more information [Line 283].

Suggested improvements

1. I suggest a deeper investigation of the enormous discrepancy in results between the two Hasler et al. datasets: relative albedo deduction versus absolute change in radiative forcing. If it's not possible to resolve this discrepancy within the scope of this manuscript, I suggest at least clearly diagnosing the reason for it and reporting it in the main text.

Thank you for this comment. Please see our responses above under the section "Data & Methodology."

2. Relatedly, I suggest a bit more description of the data and methods underlying the Hasler dataset (i.e., specific calculations, assumptions, limitations). Some description of the sources and magnitudes of uncertainty in the Hasler dataset would help the reader better evaluate this manuscript's results.

Thank you for this comment. Please see our responses above regarding additional detail we have added regarding the Hasler dataset, including magnitudes of uncertainty [Lines 279 – 294; 306 - 311].

3. I suggest more explicitly enumerating the analysis's main assumptions/priors (mentioned above):

- i. The analysis does not present a complete accounting of project non-carbon impacts. Unaccounted climate impacts could offset the reported estimates.
- ii. The analysis assumes present vs. historical tree cover for computing albedo change.

Thank you for this suggestion, which has helped us be more transparent with our previously unstated assumptions.

We have added more discussion on non-carbon impacts of ARR beyond albedo [Lines 226 - 244] and made more explicit that we focus on albedo here [Lines 13 - 15 and 246 - 258].

We additionally have specified our analysis's underlying baseline assumptions for computing albedo change [280 - 291]. For example, while true that land cover varies across time for any given location, protocols for carbon crediting with afforestation and reforestation use present or near-present (rather than historical) land cover for determining change relative to a baseline (Ellis et al, 2024), which helps avoid selection of baselines just to maximize credit production. Because we sought to integrate albedo into VCM accounting, we aligned with the methods for baseline accounting that are currently used.

4. In light of the above, and the difficulty of computing consistent, rigorous albedo deductions (as evidenced by the discrepant results between the Supporting Info and main text), I suggest a more nuanced discussion of the actual need for albedo deductions (beyond a project eligibility threshold: proposed Tiers 1 & 2) and a realistic appraisal of the challenges to reproducible implementation of albedo deductions.

Thank you for this pragmatic suggestion. We agree and have revised our recommendation, clarifying that Tiers 1 and 2 (both feasible and reproducible today) are the greatest near-term need, while both Tiers 3 and 4 would benefit from additional work [Discussions of each tier in Lines 297 – 331; call for additional research for deeper tiers in Lines 334 – 339; acknowledgement of near-term and longer-term likely adoption of tiers in Lines 352 - 363].

We have also added additional discussion (see above responses) on the need for continued scientific development to support accounting of non-albedo biophysical effects [255 - 257] and the current challenges of doing so [Lines 226 - 249].

Lastly, we note that protocols could use the existing relative albedo change dataset as a default for adopting our recommended Tiers 1 – 3. This would not preclude the use of improved albedo datasets in the future, similar to recent approaches with deforestation risk maps and leakage accounting defaults [Lines 257 - 261].

Clarity and context

The manuscript's focus on albedo is understandable, as a complete accounting of climate impacts would incorporate numerous land-atm interactions too complex/uncertain to quantify for a single reforestation site, let alone operationalize in a consistent, reproducible fashion for carbon crediting. The authors use an oft-repeated refrain that is, in fact, at odds with the reality of the carbon market.

"Participants in the carbon markets should be held accountable to the best available science without invalidating previous efforts that used the best science available at that time."

Buyers enter the market seeking to purchase real emissions reductions. An appeal to the "best available science" of the past does not rectify the loss of millions of dollars on credits rendered worthless. Buyers simply stop purchasing credits.

So the question becomes: what accounting domain is sufficiently large to ensure robust, conservative crediting in the aggregate, but sufficiently well-defined to be tractable. Tractable in the following sense:

1. All crediting calculations (or algorithms) can be validated against independent measurements to ensure consistent, reproducible crediting.
2. Uniform, conservative values are assumed for any crediting quantities for which consistent, reproducible calculation is infeasible.

I suggest the Discussion section evaluate the manuscript's proposed albedo deductions to crediting within this framework, or at least acknowledge more directly and grapple with these very real market constraints/realities.

Thank you for this thoughtful comment, which we feel has helped us reflect on how to better align our work and grapple with market realities in the following ways.

First, we have removed the notion that scientific advancements should not invalidate previous efforts and instead call for continuous onboarding of scientific advancements within market protocols [Lines 342 - 344]. This is better aligned with our intention to motivate improved scientific rigor, in line with market calls, rather than suggest that we should not rectify past accounting omissions or mistakes.

We also note that one of the benefits of the Hasler et al, 2024 dataset is that it comes from an independent third-party without incentive for particular market outcomes, making it an appropriate default dataset for the VCM [Line 366 - 369]. Our recommended tiers, particularly Tiers 1 – 3, are also reproducible and would be simple to uniformly apply across market protocols. We note that Tiers 3 and 4 especially will benefit from improved data resolution, specificity to project circumstances, dynamically updating data [Lines 203 – 205; 286 – 294; 333 – 340; 374 – 377]. In the meantime, we specify that our tiered approach provides for near-term, independent, replicable, and implementable conservativeness within the market as such work unfolds.

In addition, we more directly link and provide examples around how protocol improvements toward independent datasets, reproducibility, and conservativeness unlock trust and confidence that enables climate finance to flow toward the most effective and least harmful projects [Lines 345 – 350; 357 – 360; 365 - 378].

References

Yes, the manuscript contains appropriate references.

Your expertise

Please indicate any particular part of the manuscript, data or analyses that you feel is outside the scope of your expertise, or that you were unable to assess fully.

The manuscript uses Hasler et al. 2024 for quantification of expected albedo change due to reforestation (Hasler et al., 2024). A detailed assessment of the quality/methods underlying this dataset is beyond my expertise.

REFERENCES

Hasler, N., Williams, C. A., Denney, V. C., Ellis, P. W., Shrestha, S., Terasaki Hart, D. E., Wolff, N. H., Yeo, S., Crowther, T. W., Werden, L. K., & Cook-Patton, S. C. (2024). Accounting for albedo change to identify climate-positive tree cover restoration. *Nature Communications*, 15(1), 2275.

Hurtt, G. C., Chini, L. P., Frolking, S., Betts, R. A., Feddema, J., Fischer, G., Fisk, J. P., Hibbard, K., Houghton, R. A., Janetos, A., Jones, C. D., Kindermann, G., Kinoshita, T., Klein Goldewijk, K., Riahi, K., Shevliakova, E., Smith, S., Stehfest, E., Thomson, A., ... Wang, Y. P. (2011). Harmonization of land-use scenarios for the period 1500–2100: 600 years of global gridded annual land-use transitions, wood harvest, and resulting secondary lands. *Climatic Change*, 109(1-2), 117–161.

Hurtt, G. C., Chini, L., Sahajpal, R., Frolking, S., Bodirsky, B. L., Calvin, K., Doelman, J. C., Fisk, J., Fujimori, S., Klein Goldewijk, K., Hasegawa, T., Havlik, P., Heinemann, A., Humpenöder, F., Jungclaus, J., Kaplan, J. O., Kennedy, J., Krisztin, T., Lawrence, D., ... Zhang, X. (2020). Harmonization of global land use change and management for the period 850–2100 (LUH2) for CMIP6. *Geoscientific Model Development*, 13(11), 5425–5464.

Nobre, C. A., Sellers, P. J., & Shukla, J. (1991). Amazonian deforestation and regional climate change. *Journal of Climate*, 4(10), 957–988.

Shukla, J., Nobre, C., & Sellers, P. (1990). Amazon deforestation and climate change. *Science (New York, N.Y.)*, 247(4948), 1322–1325.

References Cited in Responses

Cabiyo, B., Field, C. B. (2025). Embracing imperfection: Carbon offset markets must learn to mitigate the risk of over-crediting. *PNAS Nexus*. <https://doi.org/10.1093/pnasnexus/pgaf091>

Ellis, P. W., Page, A. M., Wood, S., Fargione, J., Masuda, Y. J., Carrasco Denney, V., Moore, C., Kroeger, T., Griscom, B., Sanderman, J., Atleo, T., Cortez, R., Leavitt, S., Cook-Patton, S. C. (2024). The principles of natural climate solutions. *Nature Communications*. 15(1), 547–547. <https://doi.org/10.1038/s41467-023-44425-2>

Gold Standard (GS). (2024). Afforestation – Reforestation GHG Emissions Reductions & Sequestration Methodology. <https://globalgoals.goldstandard.org/403-luf-ar-methodology-ghgs-emission-reduction-and-sequestration-methodology/>

Gold Standard (GS). (2025). Gold Standard for the Global Goals: Principles and Requirements. <https://globalgoals.goldstandard.org/101-principles-requirements/>

Griscom, B. Adams, J. Ellis, P. W., Houghton, R. A., Lomax, G., Miteva, D. A., Schlesingere, W., H., David, S., Siikamäki, J. V., Smith, P., Woodbury, P., Zganjar, C., Blackman, A., Campari, J., Conant, R. T., Delgado, C., Elias, P., Gopalakrishna, T., Hamsika, M. R., Herrero, M., Kiesecker, J.,

Landis, E., Laestadius, L., Leavitt, S. M., Minnemeyer, S., Polasky, S., Potapov, P., Putz, F. E., Sanderman, J., Silvius, M., Wollenberg, E., Fargione, J. (2017). Natural climate solutions. *Proceedings of the National Academy of Sciences*. 114 (44), 11645-11650. <https://doi.org/10.1073/pnas.1710465114>

Karnik, A., Kilbride, J. B., Goodbody, T. R. H., Ross, R., Ayrey, E. (2025). An open-access database of nature-based carbon offset project boundaries. *Scientific Data*. 12(1), 581–589. <https://doi.org/10.1038/s41597-025-04868-2>

Accounting for Albedo in Carbon Market Protocols
Nature Communications submission (NCOMMS-24-69859A)
August 2025 Responses to Reviewer Comments

Reviewer comments received July 30, 2025 and responses

Reviewer 1

COMMENTS TO THE AUTHOR(S):

The authors have spent a significant amount of time revising their manuscript titled “Accounting for Albedo in Carbon Market Protocols”. The authors have addressed certain aspects of feedback, which include much more defined headings and subsections, as well as cleaner transitions and a more streamlined Result and Discussion sections. The Discussion section underlying uncertainties within the paper provides a much clearer picture of the advantages and feasibility of application of different ARR projects. Additionally, the author has made a great effort to be transparent in explaining removal of outliers, explanation of important concepts and provided a stronger foundation for their tiered approaches for current and projected albedo credits.

However, I was disappointed to see that there were other aspects of the manuscript which were not adequately updated. I believe the current literature review is still not substantial. Instead of strengthening the current introduction with the inclusion of new information, citations and references were only instead added to current literature.

Thank you for encouraging us to further engage with the literature. We are attempting to retain format-appropriate conciseness and write for our expected general audience, but agree that additional context for our study is important. We have rewritten the second paragraph of the introduction to highlight 1) that albedo effects have been known for a long time, 2) they have most commonly been identified in boreal regions, and 3) this has largely led to avoiding forestation projects for climate mitigation purposes in northern latitudes. There are a handful of other more specific studies (e.g., effects of moose browsing on albedo; proposing albedo as an ecosystem service of young plantation forests) that we feel are not appropriate for our particular study. If there were specific points you were hoping to see included, please feel free to suggest those – however, we hope that this new historical context is the additional information you are looking for. Our rewritten second paragraph is here for your convenience:

“One biophysical factor that has received renewed attention is surface albedo, or the amount of shortwave radiation reflected to space relative to what is absorbed by the surface and converted to heat in the Earth system (Figure 1). Changes in the albedo of Earth’s surface can substantially contribute to the net climate impact of forestation projects, and are a well-known phenomenon

that predates recent interests in scaling NCS^{10,11,12}. Specifically, while reforestation sequesters atmospheric CO₂ through forest growth, it also tends to decrease surface albedo, which can partially (or fully) cancel out the climate benefit of CO₂ removal^{9,12} (Figure 1). This offsetting behavior has most commonly been described for boreal regions of the globe¹⁰, which has led to general recommendations to prioritize reforestation of equatorial regions over northern latitudes¹³. However, recent advances in remote sensing and geostatistical modeling are now providing improved and spatially-resolved assessments of albedo shifts from forestation, prompting additional research and innovation in the space^{14,15}.”

Many of the sections of the manuscript which were highlighted as ‘updates/new text/explanations’ were already present before the initial review (and confirmed when revisiting the first iteration of the submitted manuscript), with very little to no improvements in context or revision.

We apologize if we made errors in where we noted new text; however, we do feel as though we have substantially updated the manuscript based on your and the other reviewers’ comments. You in fact note this above (in which you remark on the restructured, streamlined, and improved clarity of the results, discussion, and methods sections). We are unclear in terms of which sections you are referring to here, but are happy to try to improve the text if you make specific suggestions.

The authors did add a few additional sentences which discussed their focus on ARR within the Introduction, but greater care and detail is needed for expanding on exactly what are the inadequacies of current research. and where in the literature this manuscript and work is filling a knowledge gap.

Thanks for this suggestion. We have revised the second paragraph of the intro to better situate this study in the literature, as well as provide additional points to better justify its importance.

I believe the manuscript still warrants a strong merit for publication with these feedbacks more satisfactorily addressed.

Thanks for these comments. We have added additional justification to the introduction and throughout the manuscript. Regarding the knowledge gap that our study seeks to fill, we have provided an assessment of albedo effects on reforestation projects on the voluntary carbon market, which is a major channel for financing towards reforestation projects and a key mechanism for climate mitigation activity. We also propose (in detail) ways in which albedo accounting could be incorporated into carbon crediting protocols.

This has not yet been done; however, there are calls for this in the literature. For example, a study in *Nature* (out this last week; Anderegg et al 2025) has called for additional research on incorporating albedo into forest-based climate solutions (but in brief and general terms). Similarly, a country-level study (by Healey et al, 2025, *Environmental Research Letters*) was published a few months ago and performs a similar assessment for forest expansion in the United States, but does not specifically address carbon crediting projects.

We have tried to better highlight this context and the novelty of our study in the introduction, in which we have revised the 4th paragraph:

“Despite the potential for albedo benefits, albedo deductions are much more common due to the generally low albedo of forests relative to most other land cover types. Albedo deductions may even occur in tropical or temperate biomes, where forestation projects have generally been encouraged¹³. While the potential for albedo to alter the climate benefits of forest cover expansion is well known, we lack understanding of the impacts on existing NCS projects. Additionally, proposals for how albedo changes could be integrated into protocols for carbon market accounting are being actively discussed within the literature^{8,18}. While recent work has quantified shifts in albedo from forest expansion at national scales¹⁵, no such analysis has been performed for projects on the Voluntary Carbon Market (VCM), which seek to directly finance their activities by quantifying their climate impacts.”

Other:

Line 75-76: Datasets that quantify albedo effects from ARR are now available, and albedo change from ARR is expected to be consequential how? Simply listing them as a reason without a proper explanation is more sufficient. Elaborate in greater detail why ARR data is becoming more available, why is this important, and why it is becoming more mainstream over other types of nature-based solutions.

Through the edits described above, we have further elaborated on why ARR data is becoming more available (recent advances in remote sensing and geospatial modeling, Lines 52 – 54); how this is consequential and important (because ignoring albedo in NCS projects can lead to inflated estimates of climate mitigation or even projects that exacerbate rather than mitigate climate change, Lines 64 - 65); and why ARR is becoming more mainstream (the VCM is increasingly financing these activities (Lines 84 - 87) and credit buyer coalitions are increasingly seeking out ARR removals (Lines 93 - 96)).

Line 84: change ‘are’ to ‘were’.

Thank you for the edit; we have made this edit (now Line 104).

Line 226: What other biophysical factors are you referring to?

We elaborate and provide examples of these other factors in the sentences immediately following this one (Lines 250 – 253).

Line 470-471: Rephrase this sentence to better understand your project proportions.

We have rephrased this sentence structure to better articulate how we mapped projects’ projected credits to biomes using the projects’ acres in each biome (Line 482).

Reviewer 2

I have no further comments and thank the authors for their through re-working of their manuscript.

Thank you for your time and attention in reviewing our work!

Reviewer 3

The authors' have reasonably incorporated the suggestions/feedback provided by the reviewers in the form of revised text. The main analysis – extraction of a project-level albedo deduction from the Hasler et al. 2024 dataset – is straightforward.

Policy recommendations, of course, are intrinsically more subjective, and therefore need not hold up publication. The authors' Tier 1 and Tier 2 proposals – effectively an optional or mandated regional filter on siting reforestation projects to avoid large albedo reductions – is eminently sensible. Such a filter could even be achieved using a simple map of historical forest cover, particularly to avoid projects in grassland and arid biomes with little chance of durable carbon storage, possibly negative biodiversity impacts, and large albedo effects.

On the question of computing albedo deduction relative to historical or present-day albedo (most relevant for boreal reforestation), the authors simply offer the justification of conventional practice. The implication is that a forest carbon credit should be a contractual obligation encompassing not only carbon impact, but total radiative forcing impact.

A discussion of the minimum required properties of an asset for sustaining a functioning, liquid asset market is obviously beyond the scope of the manuscript. However, any asset with sufficient confidence for sustaining a market requires defining an accounting boundary within which all underlying asset calculations are standardized, reproducible, and independently verifiable (the data not simply the calculations must be verifiable). In other words, this accounting boundary defines what is computationally possible.

The authors' Tiers 3 & 4 recommendations run up against these limits. Although they recur to the phrasing that future science will improve, quantification and attribution of more atmospheric processes to individual projects (i.e., extending the asset accounting boundary to encompass radiative forcing) approaches the fundamentally unresolvable/unquantifiable. The gap lies not in the state of the science, but in the disconnect between the scale of an individual reforestation project (often on the order of a few hundred to thousand hectares) and the scale of atmospheric/radiative processes.

The forest carbon market is at present trivially small, and will never scale without a more strict definition (and calculation) of the carbon asset, if the asset remains a carbon offset meant to compensate 1:1 for buyer emissions. A paradigm shift that dispenses with the notion of an offset and transitions to a contribution or fee paid to reforestation would enable more flexible accounting of aggregate climate impact across the entire market (as opposed to individual projects).

This policy critique doesn't bear directly on the manuscript's main quantitative analysis – extracting project-level Hasler et al. albedo deductions for a large sample of existing reforestation projects. I don't wish to delay publication; I simply offer these remarks, such that they are available to the public in the supplementary “peer review file.”

Thank you for providing these additional comments, which will be helpful additions to the supplementary peer review file. We agree that there are some important policy and market considerations to be ironed out, and we hope that our suggestions and ideas provide fodder for future iteration. We discuss differences of project and atmospheric scales in lines 261-273 and 361 - 364. We are glad to see ongoing discussion of the points you suggest above and refer our readers to Anderegg et al, 2025 (cited in the manuscript) for discussion of a “contribution claims” approach.